# The turbulent future brings a breath of fresh air

Camilla W. Stjern ®[1] ✉, Øivind Hodnebrog ®[1], Gunnar Myhre ®[1] & Ignacio Pisso[2]

Ventilation of health hazardous aerosol pollution within the planetary boundary layer (PBL) – the lowest layer of the atmosphere – is dependent upon turbulent mixing, which again is closely linked to the height of the PBL. Here we show that emissions of both $CO_2$ and absorbing aerosols such as black carbon influence the number of severe air pollution episodes through impacts on turbulence and PBL height. While absorbing aerosols cause increased boundary layer stability and reduced turbulence through atmospheric heating, $CO_2$ has the opposite effect over land through surface warming. In future scenarios with increasing $CO_2$ concentrations and reduced aerosol emissions, we find that around 10% of the world's population currently living in regions with high pollution levels are likely to experience a particularly strong increase in turbulence and PBL height, and thus a reduction in intense pollution events. Our results highlight how these boundary layer processes provide an added positive impact of black carbon mitigation to human health.

Near-surface air pollution is a serious health hazard estimated to kill seven million people worldwide every year[1]. Long-term exposure is connected to increased probability of chronic asthma, pulmonary insufficiency, cardiovascular diseases, and mortality, but even short-term exposure during episodes of severely enhanced pollution is associated with increased morbidity rates related to cardiovascular diseases[2–4]. Although many regions of the world have seen a transition from historical increase to decrease in aerosol pollution over the past decades[5], aerosol exposure remains a health issue in particular in highly populated regions. However, attribution of causes into local/remote emission changes or changes in climate or weather conditions remains unclear, as exemplified by the large number of studies analyzing pollution trends in strongly polluted regions of China[6–10].

The concentration of near-surface pollutants is modified by the meteorology within the planetary boundary layer (PBL), with peak pollution events typically resulting from a combination of high emissions and unfavorable meteorological conditions such as stagnant, stable air with no precipitation[6]. Trace gases and aerosols emitted at the surface are diluted by turbulent mixing throughout the PBL[11,12]. The concentration of pollutants near the surface will thus also depend on the height of the PBL—a shallower PBL with less vertical mixing will favor stronger accumulation of pollutants near the surface[13,14]. The PBL

height is strongly linked to the boundary layer lapse rate: a weak lapse rate will be conducive to a shallower PBL, and vice versa.

Recent findings indicate the existence of a feedback effect wherein high emissions of absorbing aerosols, such as black carbon (BC), cause reduced turbulence in the lower atmosphere. This boundary layer feedback effect can exacerbate episodes of severely enhanced pollution substantially[15–18]. Absorbing aerosols initiate atmospheric heating through solar absorption and re-emission of long wave radiation. As shown in observational studies (e.g., refs. [19],[20]), the way this atmospheric heating influences the planetary boundary layer depends on the altitude of the particles. If the BC particles remain close to the surface, this added near-surface heating can increase convection and turbulence. However, the particles are often transported upwards in the atmosphere, and at higher altitudes atmospheric heating reduces the boundary layer lapse rate, resulting in a shallower PBL. Thus, under such conditions a feedback loop is initiated where more absorbing aerosols cause reduced turbulence, a narrower PBL and an increase in PBL pollution levels.

The boundary layer feedback has most often been linked either to aerosols in general[15,21], or specifically to absorbing particles[18,22,23] as exemplified above. Still, scattering aerosols such as sulfate, while having an indirect effect on atmospheric temperatures (their radiative

[1]CICERO Center for International Climate Research, Oslo, Norway. [2]Norwegian Institute for Air Research (NILU), Kjeller, Norway.
✉ e-mail: camilla.stjern@cicero.oslo.no

impact operates in the short-wave spectrum only), increased atmospheric stability by cooling the surface. In fact, scattering aerosols have been shown to increase the frequency of lower-troposphere temperature inversions over the US Southern Great Plains, although the effect was much weaker than for absorbing aerosols[24]. Moreover, emissions of greenhouse gases (such as carbon dioxide) increase the surface temperature and in turn reduce the stability in the PBL. In principle, therefore, absorbing particles are not the only anthropogenic emissions with a potential for triggering the boundary layer feedback.

In this paper, we investigate the effect of individual climate drivers on the boundary layer feedback, focusing on land regions. In contrast to previous studies, we analyze the boundary layer feedback based on multi-year simulations rather than short case studies. We use a state-of-the-art global climate model (CESM2, see Methods), with supporting simulations from a higher-resolution regional climate model (WRF, see Methods), to investigate how idealized perturbations of black carbon (BC) emissions influence turbulence and the PBL compared to perturbations of carbon dioxide ($CO_2$) and sulfate ($SO_4$). All three climate drivers will influence atmospheric stability. We hypothesize, therefore, that they all influence PBL properties and depth, and aim to see whether such PBL changes are linked to changes in turbulence, and whether there are signs of a boundary layer feedback effect in terms of changes in near-surface aerosols.

Our focus is on responses over land, as aerosols over ocean will have little health impact on people. We also perform an additional analysis centering on the eastern parts of China. This region includes the North China Plain, which is the most polluted district in China and has been studied widely due to local circulation and topography effects that tend to exacerbate haze pollution[21]. It is also a region that has experienced very strong trends in pollution in the past and that has uncertain near-term trends[25]. The region's high population density increases the potential severity of changes in near-surface aerosol concentrations.

## Results

### Global turbulence response to increased greenhouse gases and aerosols

Concentrations of both BC (Fig. 1a) and $SO_4$ (Fig. 1b) are by far highest in the regions of India and East China. This means that in our model experiments, aerosol emission perturbations will provide particularly strong impacts in these areas. The experiment involving a doubling of $CO_2$ concentration has a globally uniform perturbation. Figure 1c–n shows how the three different perturbations (a tenfold increase in BC, fivefold increase in $SO_4$ and a doubling of $CO_2$) influence near-surface-temperature, boundary layer stability, boundary layer turbulence and boundary layer height. As both the spatial and temporal resolution of CESM2-CAM6 is too coarse to resolve PBL turbulence, the Richardson number (Ri) is used as a proxy for this parameter (see Methods). Higher values of Ri indicate higher probabilities of more non-turbulent conditions, while higher values of the Brunt-Väisälä frequency indicate more stable conditions.

For the BC perturbation, we see a particularly strong increase in stability and reduction in turbulence and PBL height in the regions of high aerosol emissions. For the $SO_4$ perturbation there is also a general (but weaker) reduction in turbulence over land regions, while the $CO_2$ doubling causes an opposite reduction in stability and increase in turbulence and PBL height over land.

While our focus will be on changes over land, we still note that the BC perturbation causes similar changes to turbulence and PBL height over land and ocean, while changes to $SO_4$ emissions and $CO_2$ concentrations cause opposite changes over land and ocean. The reduction in PBL height over oceans for the $CO_2$ perturbation is established early and is visible even in an additional set of simulations where we held sea-surface temperature fixed (Fig. S1). With fixed sea-surface temperatures, the lower-tropospheric stability (as quantified by the Brunt-Väisälä frequency) is reduced over land, where surface temperatures are allowed to increase, but increases over ocean. This rapid shallowing of the marine PBL following a $CO_2$ perturbation follows from suppressed surface fluxes and reduced buoyancy[26,27], but here we show that the land/ocean pattern is opposite and is preserved even after both land and ocean temperatures are allowed to evolve for 70 years. For all three climate drivers investigated here, the long-term response in PBL height and turbulence over land is similar both in sign and magnitude to averages over the first 20 years in which sea surface temperatures are held fixed. This indicates that even for $CO_2$, the observed response over land occurs on a relatively fast time scale.

Choosing a global climate model as a tool for this study has the advantage of global spatial coverage and hundred years of temporal coverage—thus allowing for an averaging over many different seasons and meteorological background conditions. However, as mentioned, the coarse horizontal and vertical resolution means that boundary layer processes such as turbulence are parametrized rather than resolved. To investigate the consequence of model resolution to the overall response seen in Fig. 1, we perform an additional set of corresponding idealized simulations using the regional climate model WRF (see Methods), downscaled from the CESM2 fixed sea-surface temperature simulations. The WRF simulations are run at 45 km and 15 km horizontal resolution for a region over Eastern China marked in Fig. 1a. While these are still not high enough resolutions to resolve turbulence, the vertical grid spacing in WRF is ~50 m in the lowest three model layers, which means that estimates of Ri and PBL changes should be more accurate. Figure S2 shows that the diurnal evolution of the PBL height in CESM2, which has a resolution of about 100 km at these latitudes, is very similar to the WRF simulations, both in terms of baseline and changes in PBL height, albeit with a slightly delayed (1–2 h) evolution of the daytime PBL and a weaker response to the sulfate perturbation compared to WRF. Geographical patterns in baseline PBL height and PBL height changes are also robust between CESM2 and WRF, as well as between the two resolutions of WRF (Fig. S3). In summary, the regional simulations indicate that the sensitivity of our results to model resolution is relatively small, at least for the range of scales studied here, giving confidence in our CESM2 results.

Statistically significant correlations are found between changes in turbulence and changes in PBL height in Fig. 2a–c, where average monthly mean changes for each grid cell over land are shown in scatter plots for each experiment. As there have been suggestions that BC poses a greater health risk per unit mass than other aerosol species[28], we use BC concentration as a proxy for near-surface air pollution in this analysis. In Fig. 2a–c darker colors indicate grid cells with larger difference in near-surface air pollution (BC concentration) between the given perturbation run and the baseline run (see Methods). For the BC experiment in particular, regions with large near-surface aerosol changes are also associated with strong reductions in turbulence and PBL height. Note here that for the simulation involving a tenfold increase in BC, we have divided near-surface BC concentrations by ten, so that any increase in BC reflects a stronger increase than the emission perturbation itself should induce. The grid cells with the 95th percentile highest near-surface aerosol changes had a nine times higher reduction in turbulence than the average while the PBLH reduction was four times as strong. For the $SO_4$ and $CO_2$ perturbations the link between areas of turbulence changes and near-surface aerosol changes is not as strong.

Our analyses show that while the perturbations in BC and $SO_4$ in general reduce the boundary layer lapse rate, turbulence and boundary layer height, $CO_2$ tends to increase them over land. The next question is whether these changes are also associated with a change in the number of hours with elevated near-surface aerosol concentrations. Some have suggested that this positive feedback mechanism

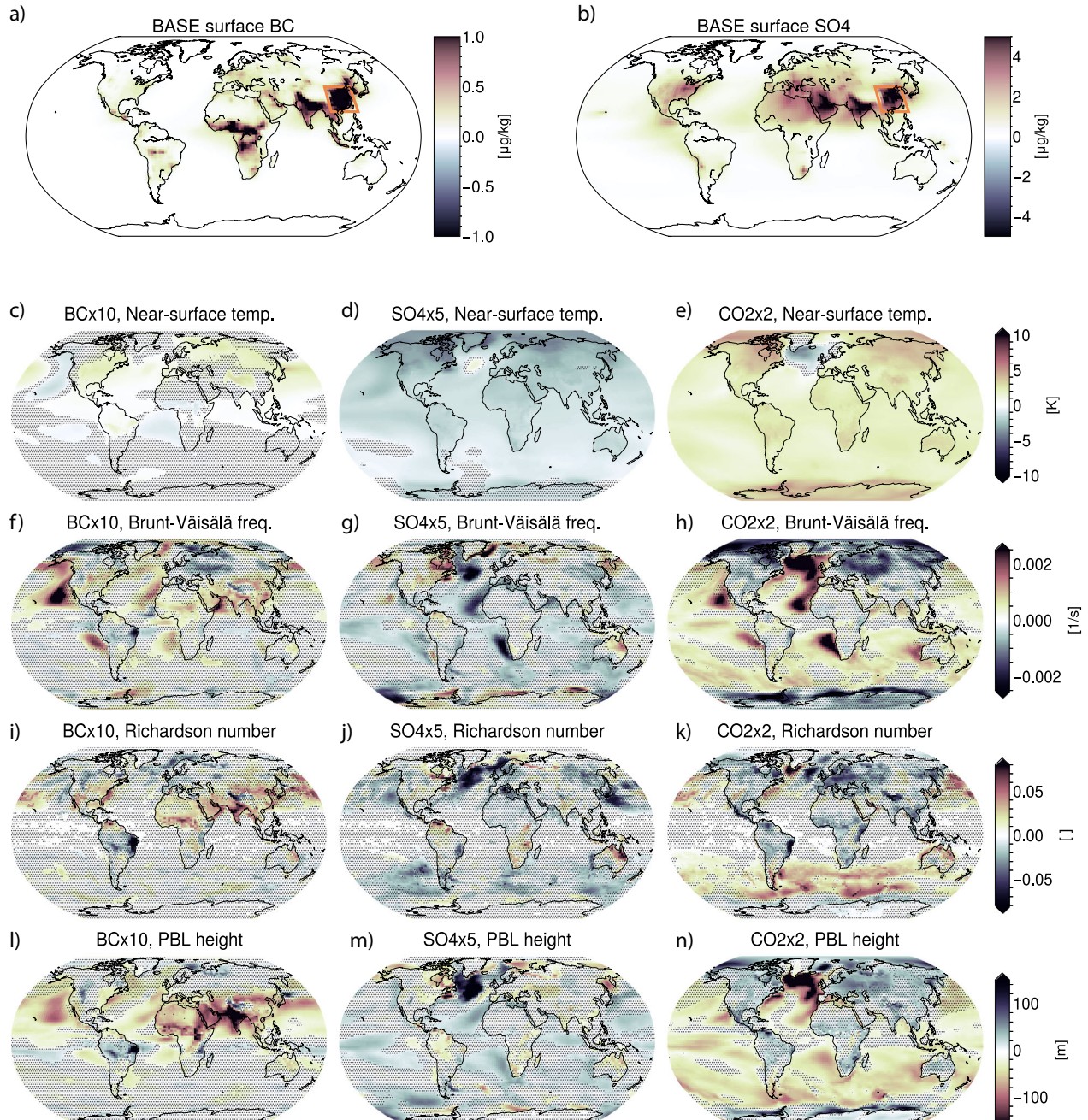

**Fig. 1 | Aerosol concentrations and global changes.** CESM2 baseline concentrations of (**a**) black carbon (BC) and (**b**) sulfate ($SO_4$). Rectangles indicate the Eastern China region. **c−n** These show changes (columns correspond to BC×10, CO2×2, and SO4×5, respectively) in near-surface temperature, Brunt-Väisälä frequency $N$ (indicating lower-atmospheric stability) at the 936 hPa level, Richardson number Ri (the ratio of turbulence-suppressing static stability to turbulence-generating vertical shear) at the 936 hPa level and planetary boundary layer (PBL) height.

only emerges under conditions that are highly polluted to begin with[15,22,23]. In the next section we therefore focus on the Eastern China region marked in Fig. 1a, and on the months from November through February as the most severe haze events typically occur in the winter season.

### Regional response in the high-emission region Eastern China

Evidence of the boundary layer feedback in China has typically been linked to absorbing aerosols[15,17,21,29,30]. For instance, based on aerosol retrievals ref. 31 found strong evidence of a boundary layer feedback in northern parts of China, where aerosols were typically mostly absorbing, while no such evidence was found in southern parts of China, where aerosols were generally less absorbing. Averaged over East China, we find that a tenfold increase in BC causes a statistically significant 7.8% reduction in annual mean PBL height. However, as we have seen above, scattering aerosols and greenhouse gases also influence the PBL height: the fivefold increase in $SO_4$ causes a 1.4% reduction, while the doubling of $CO_2$ causes an increase of 2.7%.

In Fig. 2d, we look for evidence of changes in turbulent conditions over the East China region. As mentioned, we focus on the winter (including here the months of November through February). PDFs of hourly values of the Richardson's number show that the simulation with added BC is clearly skewed towards the right, toward more non-turbulent conditions. For the simulation with added $SO_4$, there is no significant change in turbulence, while the doubled $CO_2$ simulation is skewed towards the left and thus towards more turbulence. Do these

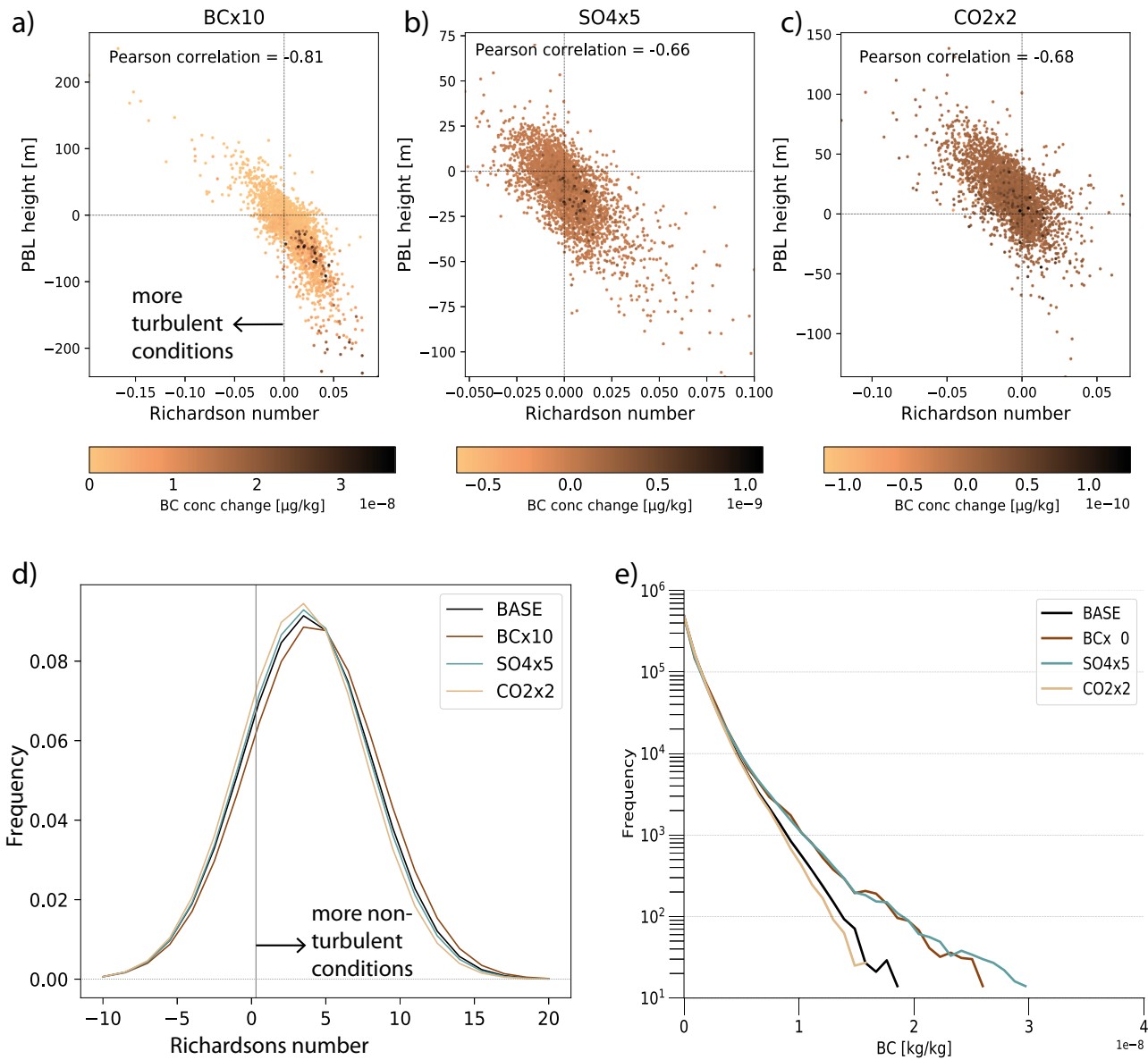

**Fig. 2 | Links between aerosols, turbulence and planetary boundary layer (PBL) height. a–c** These show scatterplots between the monthly mean change in turbulence and the change in PBL height for the three different drivers (carbon dioxide −$CO_2$; black carbon−BC; sulfate−$SO_4$), where grid cells with large BC changes are colored in darker reds (note that for the BCx10 experiments, BC values are divided by ten). Land grid cells only, all correlation coefficients are statistically significant at the 0.01 level. **d** This shows wintertime (November through February) difference in PDFs of afternoon (15.00 local time) Richardson number Ri at 912 hPa for all grid cells within the East China region. **e** Distribution of hourly values of near-surface BC concentration based on all grid cells within the East China region.

changes to turbulence and PBL height lead to changes in near-surface aerosol concentrations? Again, using BC concentration as a measure of health hazardous near-surface aerosol concentrations, we present in Fig. 2e hourly wintertime values of different levels of near-surface BC concentrations for the baseline simulation as well as the three perturbed simulations. For the most severe cases (high near-surface BC levels) the doubling of $CO_2$ causes a clear reduction, while perturbations in both BC and $SO_4$ cause an increase in the frequency of events with high near-surface pollution.

Thus, emissions in greenhouse gases as well as emissions of both absorbing and scattering aerosols seem to influence turbulence, PBL height and near-surface aerosol concentration. That does not mean, however, that it is turbulence and PBL height changes that cause the aerosol concentration changes in all cases. Many meteorological processes may influence the concentration of near-surface BC[19]. For instance, an increase in precipitation will lead to increased aerosol wet deposition and reduced aerosol numbers, and a reduction in clouds

may trigger changes in photolytically active particulate or gaseous pollutants. The nature of the simulations performed in this study does not allow for a clean separation into impacts from different meteorological effects. However, to get an indication of the processes in play and whether it is the boundary layer feedback that causes the links found above, we investigate the *timing* of the processes. We again focus on wintertime conditions over East China and take a closer look at changes in the regional mean diurnal cycles.

Crucial to the development of turbulence and the growth of the boundary layer is the vertical temperature profile. In Fig. 3a–c we show the wintertime change in the diurnal temperature profile in the lower troposphere, for the three different perturbations. The effect of BC on the temperature profile over China is striking−a strong warming around 800hPa is triggered by the absorbing aerosols which cause a strong enhancement in solar absorption at these levels. The added absorption also reduces the amount of solar radiation reaching the surface, causing a decrease in near-surface temperatures. The upper-

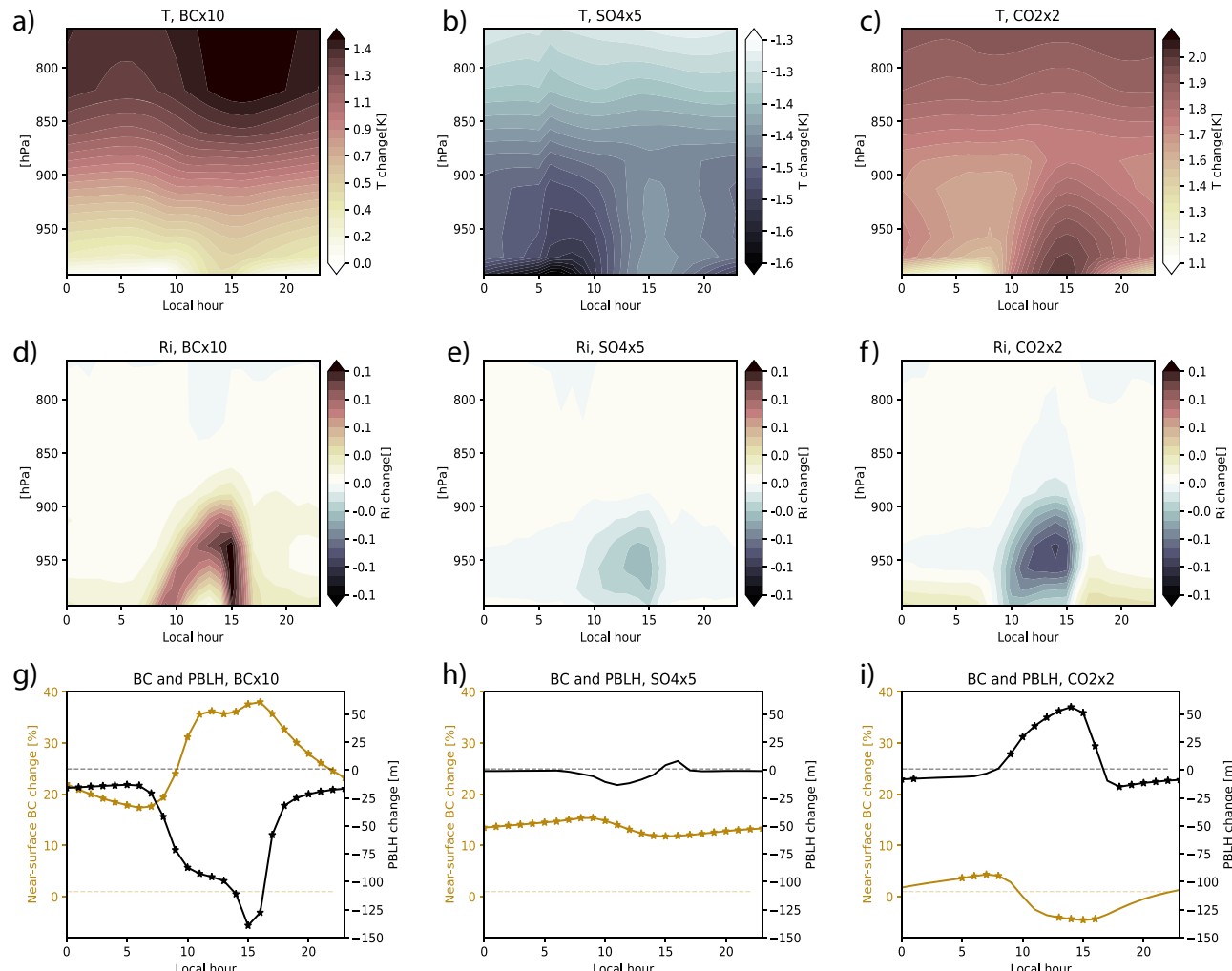

**Fig. 3 | Diurnal cycles of change.** Diurnal changes (for perturbations of black carbon BC, carbon dioxide $CO_2$ and sulfate $SO_4$) in (**a**–**c**) temperature up to 760 hPa, (**d**–**f**) Richardson number Ri up to 760 hPa and (**g**–**i**) near-surface BC concentrations and planetary boundary layer height (PBLH), in the China (CHI) region for the winter season. In (**g**–**i**) star symbols indicate hours for which changes are statistically significant by Student's $t$ test.

level warming is by far strongest in late afternoon local time, around 15.00 h. This is consistent with ref. 22, who found for megacities in China that upper-PBL heating by BC was strongest during late afternoon. These aerosol-radiation interactions cause an increase in atmospheric stability by reducing the temperature difference between the surface and higher atmospheric levels. Through reduced surface fluxes, this leads to an increase in the Richardson number, strongest around 15.00 h (Fig. 3d) but statistically significant by Student's $t$ test for all hours of the day. The diurnal variation in the BC-induced temperature increase and turbulence reduction connects closely to a change in PBL height (black line in Fig. 3g), where hours with significant change are marked with star symbols. In the BC perturbation, the PBL height change is significant for all hours of the day but strongest at 15.00 h. Finally, the increase in near-surface BC concentrations (yellow line) also shows a similar but opposite diurnal cycle, indicating a link between the processes driving the changes to turbulence and PBL height and the exacerbated near-surface aerosol conditions.

In the case of the simulation with perturbed $SO_4$, we also see an increase in atmospheric stability, this time driven by a reduction in surface temperatures as the scattering aerosols efficiently reflect a portion of the incoming solar radiation back to space. However, neither the change in turbulence (Fig. 3e) nor the change in PBL height (Fig. 3h) is statistically significant for any of the 24 h. While there is a significant increase in near-surface BC, shown also in

Fig. 2e, in this case this is clearly not driven by the boundary layer feedback. Instead, by looking at changes in the model's different aerosol modes (the primary carbon mode and the BC accumulation mode), we find that the added $SO_4$ causes a very strong reduction in primary mode and increase in accumulation mode BC. In the CAM6 MAM4 aerosol scheme, BC particles "grow" by coating by $SO_4$ or SOA, and this constitutes the strong removal of primary mode and addition of accumulation mode BC aerosols. This model response, although approximating a similar effect in nature, makes it difficult to draw conclusions as to the boundary layer feedback effect of $SO_4$ from these simulations. In a future study we will use a regional model to isolate the link more properly between the stabilizing effect of $SO_4$ and surface pollution, comparing the effects of $SO_4$ and BC.

The doubling of $CO_2$ has an influence on the atmospheric temperature profile that is opposite to the effects of BC and $SO_4$; the increase in surface temperature (Fig. 3c) makes the atmosphere less stable, and as a result we see a strong mid-day reduction in the Richardson number in Fig. 3f. The increased turbulence causes an increase in PBL height that is statistically significant at the same time of day that we see a reduction in near-surface BC particles (Fig. 3i). These changes are not as strong as for a tenfold increase in BC, and changes to atmospheric stability, turbulence, PBLH and near-surface aerosol concentration are slightly opposite during night-time. Still, on average,

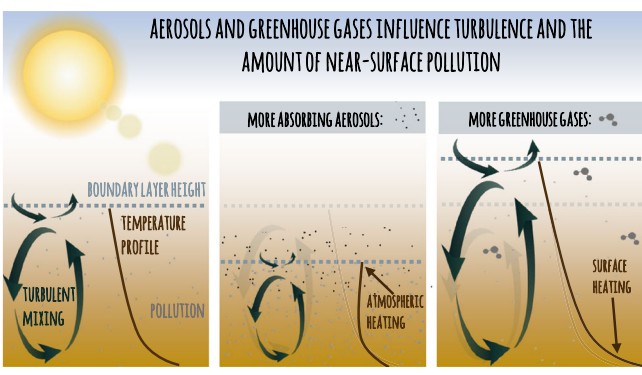

**Fig. 4 | The link between turbulence and pollution.** This schematic illustrates how absorbing aerosols and greenhouse gases can influence turbulence and through that the amount of near-surface pollution.

as seen in Fig. 2e, increased $CO_2$ leads to a reduction in near-surface aerosols.

To summarize, the link between turbulence, boundary layer height and near-surface air pollution is illustrated in the leftmost panel of the schematic in Fig. 4. The following two panels show how this is influenced by absorbing aerosols and by greenhouse gases, respectively.

The highly idealized nature and the exaggerated magnitude of these perturbations, while necessary to produce strong enough signal-to-noise ratio to analyze these processes, preclude any direct linking between our findings and current trends in $CO_2$, BC, or $SO_4$ in the real world. However, knowing how changes in these constituents will influence near-surface pollution levels can help us interpret more realistic simulations of future emission changes. In the next section, we take a closer look at such simulations from the Sixth Coupled Model Intercomparison Project (CMIP6, see Methods).

### Application to historical and future trends

Historical changes in aerosol and greenhouse gas emissions have been substantial, but finding the link between emission changes and observed historical trends in PBL height[31–34] is challenging—not only because so many other factors influence PBL height, but also because the observations are notoriously difficult to compare due to differences in data sources and PBL height approximation methods. There are several indications, however, that changes in observed PBL height are connected to emission changes. For instance, ref. 33 investigated a larger observational data set for China in general, and found an increase in PBL height from 1979 to 2004, but a robust reversal of the trend from 2004 to 2016. They found that an increase in urbanization and BC emissions since the turn of the century are strong contributors to this reversal in the PBL height trend. Ref. 32 analyzed observed trends in PBL height and also found an increase in major cities in China, reporting trends in aerosol emissions as an important co-driver of PBLH evolution. Recently, ref. 35 analyzed global PBLH-trends between 1979 and 2019 using ERA5 and MERRA2 reanalysis data, and found relatively strong discrepancies regionally between the two. A comparison to historical simulations in the latest CMIP6 ensemble found stronger agreement to MERRA2 than to ERA5, but all three data sets displayed declining PBL height over India.

In Fig. 5a we have calculated the CMIP6 ensemble mean (see Table S2 for included models) linear trend in PBL height over the entire historical period (1850–2014), for which both greenhouse gases and aerosol emissions have gone up. We find declining PBL height coincident with regions of strong aerosol emission increases, suggesting that the aerosol influence on PBL height dominates the response. Based on population data from the Socioeconomic Data and Applications Center (SEDAC)[36] we find that 83% of the world's population (in

year 2010) live in areas that have seen a reduction in the PBL height over the historical period. The response in the CMIP6 ensemble is comparable to that of CESM2-CAM6, as seen in Fig. S4.

Aerosol emission changes are estimated to continue into the future, with particularly large and uncertain trends in Asia[25]. The Shared Socioeconomic Pathways (SSP) used by the CMIP6 community[37,38] span a range of narratives of possible futures—from a collaborative world with strong international cooperation and rapid climate mitigations (e.g., SSP126) to more conflict-ridden societies with where mitigation is not a high priority (e.g., SSP370). The long lifetime of $CO_2$ precludes any substantial reductions in the atmospheric $CO_2$ concentration by year 2100 regardless of emission pathway[39]. Aerosol emissions, however, are set to go down in all versions of the future, although some scenarios (in particular SSP370) have a delayed onset of the emission reductions. Given the tendency of BC to suppress turbulence and $CO_2$ to enhance turbulence, as found in our idealized model simulations, both an increase in $CO_2$ concentration and a reduction in BC concentrations will contribute to enhanced turbulence, with the potential positive consequence of an alleviation in the number of haze events. Note that while the future reductions in BC emissions themselves will of course be the main determinant of health-related exposure to these aerosols, we are here interested in the added positive or negative influence by an associated change in PBL height. We now investigate the link between emissions and PBL height in future scenarios, to get insight into whether the boundary-layer feedback could have detrimental or advantageous impacts on near-surface pollution in the future.

In Fig. 5b, c, we show future BC emission changes for two selected SSPs; in SSP370 aerosol emissions keep increasing until around 2040–50 and decline thenceforth, while in SSP585 emissions decline from the beginning. A version of Fig. 5b, c showing also SSP119 and SSP245 is shown in Fig. S5. Figure 5d–g shows corresponding near-term (2015–2045) and long-term (2015–2100) trends in PBL height for the same two scenarios (see Fig. S6 for all SSPs). The effect of continued increases in BC emissions in the South Asian (predominantly India) region in SSP370 is clearly reflected in a continued regional reduction in PBL height in both near- and long-term. In contrast, the immediate emission reductions in SSP585 can be seen as an increase in PBL height which is strong over the Asian region but in general increases over all land regions. In SSP119, which has the strongest near-term emission reductions in BC (Fig. S5), even the near-term PBL height increases substantially over land (Fig. S6).

Although depending on the exact emission pathway of both greenhouse gases and aerosols, land regions are very likely to see an increase in PBL height in the future. If we follow the "middle of the road" pathway of SSP245, around 60% of the world's population at the end of the century (year 2100) will be living in regions where the PBL height has increased since present-day. Around 6% (more than 400 million people) will be living in regions where BC concentrations were very high (above the 95th percentile) around year 2000 and where the increase in PBL height is set to be particularly strong (above the 90th percentile). In SSP370 this number is around 3%, in SSP585 it is around 13%, and in SSP119 it is more than 30% of the world's population. Thus, in all future scenarios, though to varying extent, high-exposure regions —and the hundreds of millions of people living there—will experience the added positive impact of an alleviated boundary layer feedback effect on top of decreasing aerosol emissions. Recent evidence points to beginning reductions in aerosol emissions in the East China region. For instance, observations of BC concentrations in Beijing show a 67% reduction from 2012 to 2020[7]. Emission changes in the following century are therefore likely to involve a continued increase (or very weak reduction) in $CO_2$ concentration in combination with reduced BC concentrations. As we see here, both of these changes have the potential to spur an increase in boundary layer turbulence and boundary layer height. Hopefully, the problems of aerosol pollution

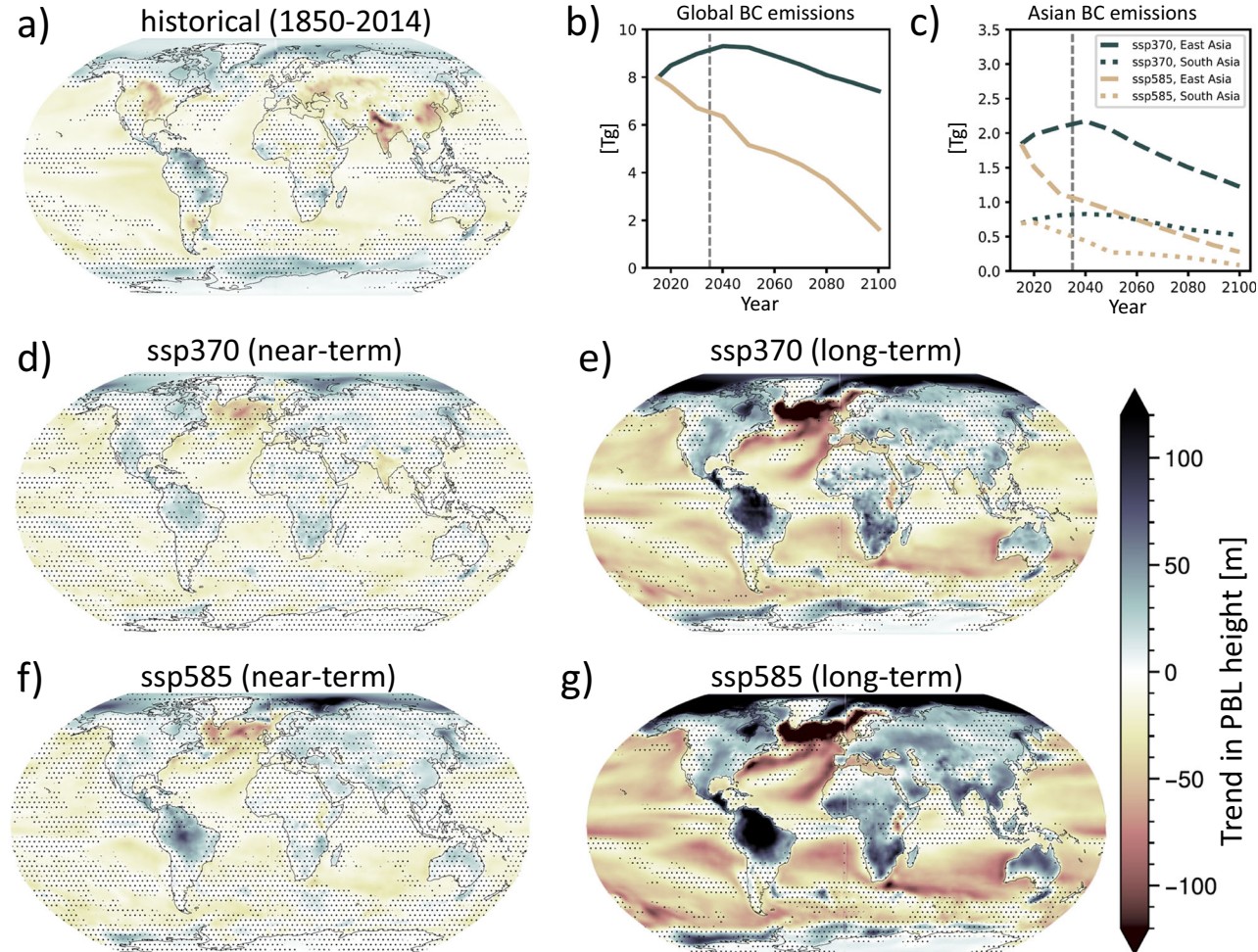

**Fig. 5 | Historical and future planetary boundary layer height (PBLH) changes.** **a** Change in PBLH over the historical period (linear trend from 1850 to 2014). **b** Global and **c** Asian emissions of black carbon (BC) in different shared socioeconomic pathways (SSPs): SSP370 and SSP585. **d–g** Near-term (2015–2045) and long-term (2015–2100) trends in boundary layer height in SSP370 and SSP585. Hatching where less than 75% of the models agree on the sign of the trend.

will eventually be alleviated in all regions of the globe, at which point the added impact of changes in turbulence is of little importance. In the meantime, air pollution is estimated to cause millions of deaths every year[1], and mitigations in the future do not necessarily provide instant relief as the number of people exposed to pollution-driven health risk is a function not only of emission levels, but also of demographic and socioeconomic factors. For instance, ref. 40 find that excess deaths from PM2.5 in India are expected to increase towards 2100, due to demographic transitions involving population growth as well as a shift towards an older population. Ultimately, exposure to air pollution depends on a myriad of factors and is likely to continue to be an issue even as emissions go down. The way anthropogenic emissions interact with turbulence and feed back on the level of near-surface air pollution is one such factor, and one that is likely to contribute with a positive sign in the future.

## Methods
### CESM2-CAM6 simulations
The present study uses the Community Earth System Model 2 (CESM2), with the Community Atmosphere Model Version 6 (CAM6) as the atmospheric component[41]. We do simulations both in a setup with fixed sea surface temperatures (fsst) as well as in a fully coupled mode (cpl), where the atmosphere component is coupled to ocean, land, and cryosphere models. CESM2's ocean component is based on the Parallel Ocean Program Version 2[42]. The fsst simulations are

run with CAM6 at a 0.9 × 1.25 finite volume grid and the cpl simulations with CAM6 at a 1.9 × 2.5 finite volume grid−both simulation set-ups using 32 vertical levels (top level at 2.25 hPa). CAM6 uses the updated Morrison-Gettelman cloud microphysics scheme[43], as well as the turbulence scheme Cloud Layers Unified by Binormals (CLUBB)[44]. CLUBB represents moist turbulence by use of a multivariate binormal probability density function, describing subgrid-scale variability in temperature, vertical velocity, and humidity. Aerosol treatments are according to the Model Aerosol Model version 4[45]. The vertical distribution of BC, relevant to the distribution of atmospheric heating and thus to stability changes, is simulated by the model and depends on processes such as dry and wet deposition, advection, and vertical transport. In a thorough observational comparison with an earlier version of CAM, the model was found to capture the main features of the variation of BC with height, though with a tendency to overestimate BC concentrations in the upper troposphere, and underestimate observed BC concentrations at lower altitudes in high northern hemisphere latitudes[45]. In China, the focus region of this study, models in general have been shown to have too low total column aerosol loads[46] and too low surface aerosol concentrations[47]. These are biases that are common to many global climate models[48], and are at least partly linked to biases in wet deposition, vertical transport, and lacking emission sources. In the present study a low bias in the intense pollution events may cause us to underestimate the boundary layer effect of BC changes.

All simulations are summarized in Table S1. The baseline simulations (BASE) represent recent conditions, with cyclic monthly climatology fixed at year 2000 for the fsst run and evolving freely for year 2000 conditions for the cpl run. Emissions of precursor gases and aerosols, which follow CMIP6[49], are also set to year 2000 (maps of baseline concentrations in BC and $SO_4$ are shown in Fig. 1a,b). Relative to the baseline, we then perform experiments with doubled concentrations of $CO_2$ (CO2x2), a tenfold increase in anthropogenic emissions of black carbon (BCx10) and a fivefold increase in anthropogenic emissions of sulfate (SO4x5). This means that in the aerosol experiments, the geographical pattern of emissions is preserved. The magnitudes of the experiments are artificially strong compared to real-world emission changes in order to produce clear enough signals in the resulting atmospheric process changes. In the perturbation experiments we use the same climatology as BASE, and $CO_2$/BC/$SO_4$ are perturbed instantaneously from the first time step. All fsst simulations are 15 years long, and the last 10 years of data were used in the analyses. The cpl simulations were started at year 2000 conditions, based on input data from a transient simulation run from 1850 to 2000. We first ran 60 years to stabilize the near-surface temperature, and until the yearly variation in the global mean temperature was comparable to the variation in CESM2 piControl simulations from CMIP6. Both baseline and perturbed experiments were then branched off from this point, and run for 50 additional years, allowing the temperature to respond to the perturbations. Finally, we ran another 20 years, for which data were analyzed.

Characterization and quantification of the turbulent behavior of the atmosphere in response to the different climate driver perturbations are based on hourly model outputs, as opposed to the default monthly mean output from the model. We calculate Brunt-Väisälä frequencies $N$ (Eq. (1) below) to quantify atmospheric stability, and Richardson numbers $R_i$ (Eq. (2) below) as a proxy of turbulence. While $N$ is a valuable quantity to characterize the atmospheric stability, $R_i$ is used to identify layers in the atmosphere that are potentially turbulent[50].

$$N = \sqrt{\frac{g}{\theta}\frac{d\theta}{dz}} \qquad (1)$$

Here, $g$ is the acceleration by gravity, $\Theta$ is the ambient virtual potential temperature and $z$ is height.

$$R_i = \frac{N^2}{(dv/dz)^2} \qquad (2)$$

$v$ is the horizontal wind speed in the above equation. The calculations were performed using the rigrad bruntv function implemented in ncl 6.4.

The Richardson number describes the relationship between static stability and vertical wind shear. Typically, an $R_i$ below a certain critical value corresponds to situations with dynamic instability and growth of turbulent conditions. Conversely, $R_i$ values above the critical value speak of weak or decaying turbulence or entirely non-turbulent conditions. Traditionally the critical $R_i$ value has been set to 0.25, but in reality it depends on various conditions and is more accurately set to a span between 0.2 and 0.5[51]. In CAM6, calculations of boundary layer height are based on the model's own calculations of $R_i$ and defined as the height of the first layer where (vertically) the $R_i$ exceeds a predefined $R_{i-critical}$ of 0.3. We note, as a caveat, that global climate models in general struggle to simulate very shallow stable boundary layers[52,53]. In this analysis, this tendency may lead to underestimates of the tendency of the boundary layer to narrow in response to increased BC.

In this work we have chosen to use the concentration of black carbon as a proxy for "near surface aerosol concentrations". This is partly motivated by the fact that these particles are shown to have a

particularly detrimental effect on health[28], and partly necessitated by the limitation of hourly output from the model. BC concentration was available as output for hourly, daily, and monthly temporal resolutions from all of the simulations. In reality, the perturbations performed in our simulations may influence the near-surface concentration of all types of pollutants. Changes in processes that only influence highly hydrophilic aerosols, or gaseous pollutants, will not be captured by our approach. However, our assumption is that the changes we see in planetary boundary layer height will cause a concentration or dilution of all types of particulate pollution and will therefore also be seen in changes in our BC output.

## CMIP6

In addition to our separate driver specific CESM2 simulations, we also look at historical and future changes in boundary layer height by available CMIP6 models. Table S2 shows a list of models included for the different CMIP6 experiments analyzed. Trends for the historical period were calculated as linear trends from 1850 to 2014, while near-term and future trends were calculated for 2015–2045 and 2015–2100, respectively.

## WRF simulations

Finally, in order to investigate the robustness of the boundary layer changes in CESM2, we have performed a separate set of corresponding idealized simulations with the regional climate model Weather Research and Forecasting (WRF)[54] version 4.3.3. The model was set up with two domains, the first covering Southeast Asia (lat/lon extent of 8.2°S-57.8°N/51.6°E-168.4°E) at a horizontal resolution of 45 km × 45 km, and the second covering East China (lat/lon extent of 17.6°N-47.7°N/95.0°E-136.2°E) at 15 km x 15 km. One-way nesting was applied and both domains used 50 vertical layers extending from the surface and up to 100 hPa, whereof 10 layers were within the lowermost kilometer. An outer boundary of 10 grid boxes was ignored in the analysis to reduce influences of boundary conditions on the results.

The WRF simulations were downscaled from CESM2 by using meteorological initial and boundary conditions from the fsst simulations as input to WRF for each of the experiments BASE, BCx10 and SO4x5. The sea-surface temperature fields (fixed for year 2000 climatology) were the same in CESM2 and WRF, and boundary conditions were updated every 6 h. In the outer WRF domain, spectral nudging of horizontal winds was applied to the CESM2 data. The WRF model was run without interactive chemistry but used monthly 3-D aerosol optical depth fields for each of black carbon, organic matter, sulfate, dust and sea salt from the CESM2 simulations in the radiation scheme in WRF, to represent the effect of aerosols on radiation. For this purpose, the Rapid Radiative Transfer Model[55] was used as the WRF radiative transfer scheme with the aer_opt=1 option activated, and with small modifications to the aerosol scheme to allow for a different horizontal and vertical resolution of the aerosol fields from CESM2 compared to the default aerosol fields. In addition, the default single-scattering albedo values for BC in WRF were considered too low and therefore changed to the default values in the CAM radiation scheme[56] for each spectral band. Other physics schemes used in WRF were the WSM 6-class graupel scheme[57] for microphysics parameterization, the Grell–Freitas ensemble scheme[58] for cumulus parameterization, the Mellor–Yamada–Janjic TKE scheme[59,60] for boundary layer parameterization, the Monin–Obukhov (Janjic)[59,61–63] surface layer scheme, and the Unified Noah land-surface model[64].

## Data availability

CESM2 results are available from download at the repository https://archive.sigma2.no/, located at https://doi.org/10.11582/2023.00021[65]. The CMIP6 data are available at https://esgf-node.llnl.gov/search/cmip6/. Population data were downloaded from the NASA Socio-economic Data and Applications Center (SEDAC), accessed on

02.06.2022: https://sedac.ciesin.columbia.edu/data/set/popdynamics-1-8th-pop-base-year-projection-ssp-2000-2100-rev01/data-download.

## Code availability
Python code for analysis of CESM2 results is available from download at the same repository as the data (https://archive.sigma2.no/, located at https://doi.org/10.11582/2023.00021[65]).

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

## Acknowledgements

All coauthors have received support from the project "GREenhouse gases, Aerosols and lower atmospheric Turbulence" (GREAT), funded by the Research Council of Norway (grant no. 275589). The computations/simulations were performed using the NN9188K project account and data was stored and shared on project account NS9188K on resources provided by UNINETT Sigma2—the National Infrastructure for High Performance Computing and Data Storage in Norway. We acknowledge the World Climate Research Programme, which, through its Working Group on Coupled Modelling, coordinated and promoted CMIP. We thank the climate modelling groups for producing and making available their model output, the Earth System Grid Federation (ESGF) for archiving the data and providing access, and the multiple funding agencies who support CMIP and ESGF.

## Author contributions

C.W.S. contributed analyses and writing, as well as model simulations. Ø.H. contributed with planning, model set-up, discussions and manuscript edits. I.P. contributed the calculated Brunt-Väisälä and Richardon number values, discussion and manuscript edits. G.M. contributed with planning, discussions and manuscript edits.

## Competing interests

The authors declare no competing interests.
