## [Peer Review File · Nature Communications]

The turbulent future – a breath of fresh airREVIEWER COMMENTS

Reviewer #1 (Remarks to the Author):

This study uses idealized simulations to indicate that future increase of CO₂ and reduction of aerosols may decrease surface stability and increase boundary layer height, which serves to reduce the concentration of pollutants at surface. While the results are useful in assessing future pollution trends, I find the study lack of novelty and the analysis is too simple. I therefore do not recommend publication of the article in NC.

Major comments:

1. The main conclusion of the study is that future reduction of BC and other aerosols and increase of CO₂ both contribute to increase surface temperature and causes more turbulence in the lower atmosphere. This is not a new finding. Many previous works, as cited in the main text, have documented the aerosol-boundary layer feedback. The role of CO₂ increase in changing boundary layer height is indeed less investigated. However, the conclusion that increased CO₂ increases surface temperature and thus decreases stability is not surprising, and the mechanism is not new. Moreover, the change in PBLH depends on the temperature profile rather than surface temperature alone. As seen in Figure 3a, CO₂ induces substantial warming in the upper atmosphere. Only between 10 and 16h there is stronger warming at surface. Therefore, it is hard to conclude that increase in CO₂ would reduce pollution.
2. The model resolution is too coarse to resolve any boundary layer process. CESM only has 2-3 levels within the PBLH, whereas to resolve turbulence, it typically requires vertical resolutions less than 50m. Therefore, this model can be used to diagnose convective stability at best, and the PBLH indicated is only convective boundary layer height. The results do not reflect any information of turbulence.
3. The analysis is very simple and many conclusions are speculative. All quantitative conclusions are based on idealized and sometimes extreme cases such 10 times BC and 2 times CO₂. The change in surface pollutants is actually the change in BC, which also largely depends on the assumed emission. Also, the changes in other pollutants, such as secondary aerosols and gases, depends on many meteorological factors beyond PBLH. These limitations should be explicitly discussed. Also, the results largely depend on the vertical distribution of the forcing agents, especially aerosols, yet there is not description how these profiles are specified in the simulation, and whether they represent real conditions.

Reviewer #2 (Remarks to the Author):

This study focuses on the interactions between radiative forcers, specifically black carbon and sulphate aerosols and carbon dioxide (CO₂) and their potential to influence planetary boundary layer height through perturbing the temperature profile of the PBL and influencing turbulent motion. This study fits well and contributes to an increasing number of studies focusing on the aerosol-PBL feedback loop. This study is novel in the aspect of the use of a climate model rather than a smaller scale model which directly resolves turbulent motion. The advantage of using this type of model comes with the ability to assess the impact of various scenarios into the future to understand the impact changes in emissions could have on pollution episodes. The manuscript is overall well written and the methodology is clear. General comments and suggestions to the authors are detailed below.

General Comments:

- 1) Throughout the manuscript it is stated that black carbon can only cause suppression in the PBL, through reducing solar radiation reaching the surface. However, this is dependent on the altitude of the BC layer. A few papers including Wang et. al 2018 (<https://doi.org/10.5194/acp-18-2821-2018>) and Slater et. al 2022 (<https://doi.org/10.5194/acp-22-2937-2022>) highlight the impact that the altitude of the BC layer can have on a) The magnitude of suppression caused by BC in the PBL and b) Whether it has the potential to promote turbulence e.g if a BC layer exists only at the surface and not throughout the atmospheric column. Whether these conditions occur often is uncertain but some studies have shown there can be variation in both concentrations and

compositions of aerosols across the column, particularly in very polluted conditions. It may be worth adding a sentence or two to highlight this - both in relation to BC and SO₄.

2) An important output of this study could be the links between climate change mitigation (e.g. reducing emissions of CO₂ and BC) and the additional benefits for reduced pollution and consequently human health beyond those which are typically cited. E.g. reducing emissions of CO₂ and BC could have the added impact of allowing for more vertical mixing/turbulence and therefore have an additional effect (beyond reducing emissions) on human health through decreasing the number of severe pollution episodes (both in longevity and severity)

3) Relating to point 1 - does the model simulation set up here assume uniform vertical aerosol profiles? It would be useful to state this somewhere and to highlight it as a potential limitation as well as further highlighting the limitations of using a coarse model to understand PBL turbulence.

4) Line 145-150: It is stated that in Eastern China the PBL feedback effect is larger due to increased absorbing aerosols compared to in Southern China. However, there may be other factors influencing this - for example meteorological conditions which have been shown to impact this effect see Wang et al. 2019 (<https://doi.org/10.5194/acp-19-6949-2019>).

5) Does the model take into account the impacts of projected increases in surface temperature due to climate change? What impact does increased surface temperatures have on this effect?

Throughout the manuscript it would be useful to clarify a few terms:

Line 24: Morbidity rates

Line 34: What is meant by unfavourable meteorological conditions e.g. prevailing winds, pressure systems, temperature, humidity etc.

Reviewer #3 (Remarks to the Author):

Reviewer comments for the paper "The turbulent future – a breath of fresh air" by Stjern et al., submitted for publication in Nature Communications

The authors utilize data from global model and from CMIP6 ensemble to explore the impacts of feedback mechanics impacting the boundary layer height and turbulence. The driving force of these changes stem from changes in aerosol emissions and anticipated CO₂ concentrations in future climate. The noteworthy analysis includes the use of global data set from model ensemble to explore the impacts of concurrent changes in aerosol emissions and warming climate and turbulence. The authors depict a convincing but a hopeful scenario that could improve air quality in rapidly urbanizing environment in the coming decades, aided by strong air quality controls that would also improve the health of the urban population.

As a summary, the work is a relevant contribution providing novel insights into aerosol-boundary layer feedbacks. Therefore, I recommend to publish the paper after considering my comments that I summarize in the comments below.

Significance of the work to the field, originality

The authors utilize a global data set in their analysis of impacts of changing emissions and boundary layer height and turbulence. As the authors point out, these feedback mechanisms are typically only addressed in short experiments and not within a global context in the changing climate. This is the novelty of the work.

The authors differentiate absorbing aerosols and scattering aerosols in their analysis in a common framework. This is an important step forward in the analysis of boundary layer – aerosol feedback analysis.

Solidity of the conclusions and claims, any additional evidence needed?

However, since I'm not a global modeler, I cannot address fully the capacity of the models to perform well enough for such endeavors. I list below few concerns that should be addressed and clarified:

1) The simulations with CESM2-CAM6 are performed with doubled CO₂ and separate model runs with tenfold increase in BC, and fivefold increase in SO₄. Is this done uniformly for the global emissions with keeping the regional differences? Are these very high increases made to enhance the anticipated signal? Are there any unwanted consequences (or interactions) coming from the perturbations? How does e.g the surface coating of BC due to sulfuric acid condensation influence the overall picture?

How reasonable is the tenfold and fivefold increase in BC and SO₄ emissions? On one hand, according to Xu et al. (2021) the BC has approximately doubled from 4 Tg to 8 Tg in 57 years. 10-fold increase is large in this context. On the other hand, global sulfur (as SO₂ and as SO₄) emissions are declining (Aas et al. 2019).

2) Is the vertical resolution of the model enough to capture the full dynamics of the growing boundary layer during its diurnal cycle? What are the consequences of using Ri as an indication for the turbulence? Can these be tested against observations or against higher resolution models?

3) How about horizontal resolution? How are the differences in local anthropogenic heating (urban heat island) taken into account in the development of turbulent in urban environments?

Flaws in the data analysis, interpretation and conclusions. Revisions needed?

I don't see any flaws in the analysis or interpretation of the results. However, see comments above regarding the model resolution and comment on the perturbations.

Soundness of methodology, does it meet the expectations

The methodology fits the work, given that the model spatial and temporal resolution are adequate. The work provides as a step forward in global context by expanding the process level studies and regional studies referred in the paper.

Enough details provided?

Generally yes, but technical details regarding the modeling should be elaborated in more detail.

Detailed comments.

The paper is very well written. For the internal structure, I actually got more interested in the topic with the detailed analysis and Figure 2. The story in the paper could improve, if it was approached from the details into the global picture.

Line 252-253: Can you quantify the reduction of intense haze events?

In superlatives, please use "the" in proper places.

References

Aas et al. (2019) Global and regional trends of atmospheric sulfur, *Scientific Reports* 9, 953.

Xu et al. (2021) Updated Global Black Carbon Emissions from 1960 to 2017: Improvements, Trends, and Drivers, *Environ. Sci. Technol.* 2021, 55, 12, 7869–7879.

Response to reviewers

“The turbulent future – a breath of fresh air”, by Stjern et al.

Reviewer #1 (Comments to the Author):

This study uses idealized simulations to indicate that future increase of CO₂ and reduction of aerosols may decrease surface stability and increase boundary layer height, which serves to reduce the concentration of pollutants at surface. While the results are useful in assessing future pollution trends, I find the study lack of novelty and the analysis is too simple. I therefore do not recommend publication of the article in NC.

We thank the reviewer for this input. We note that the two other reviewers have emphasized the novelty of our study. The study builds on well-known physical processes (e.g. Wilcox et al., 2016) that have earlier been shown in observations and high-resolution models (e.g. Senf et al., 2021). Here we show that these exists on a global scale and that future emission pathways of aerosols and CO₂ are likely to increase turbulence and lead to less pollution. We have performed a set of new model simulations which show that the processes found in the global climate model are similar to those in model simulations with much higher resolution. We have also taken care to improve descriptions where needed, and to discuss caveats (e.g., the use of BC as a proxy for near-surface aerosol pollution) and their potential impacts on our results.

Major comments:

1. The main conclusion of the study is that future reduction of BC and other aerosols and increase of CO₂ both contribute to increase surface temperature and causes more turbulence in the lower atmosphere. This is not a new finding. Many previous works, as cited in the main text, have documented the aerosol-boundary layer feedback.

The reviewer is of course correct, this manuscript does not present findings that demonstrate formerly unknown physical mechanisms in the atmosphere. But while observational studies do indeed exhibit a more valid representation of real-world processes than the global climate model, such studies are limited to a single city or region over a limited time period. Although the tools chosen for this study have limitations, they have the advantage of allowing for a comparison between the effects of different climate drivers at a global scale and for a multiple year time scale. Such a comparison has not been shown before in the context of the aerosol-boundary layer feedback. Thus, our finding that CO₂ causes an increase in turbulence over land on a global scale is novel.

The role of CO₂ increase in changing boundary layer height is indeed less investigated. However, the conclusion that increased CO₂ increases surface temperature and thus decreases stability is not surprising, and the mechanism is not new.

Indeed, the fact that CO₂ destabilizes the lower atmosphere is no new finding. However, the context of how its influence on the boundary layer height plays a role in near-surface aerosol conditions is new. The comparison between the boundary layer effect of CO₂ and the corresponding effect of other anthropogenic emissions (BC and SO₄) is also new. Thus, while the mechanisms we show are neither surprising nor new, we believe that the perspective is.

Moreover, the change in PBLH depends on the temperature profile rather than surface temperature alone. As seen in Figure 3a, CO₂ induces substantial warming in the upper atmosphere. Only

between 10 and 16h there is stronger warming at surface. Therefore, it is hard to conclude that increase in CO₂ would reduce pollution.

Absolutely, it is the change in temperature profile (and the stability change it induces) and not the surface temperature change alone that is important. Figure 3a does indeed indicate reduced stability only in the daytime hours (when the surface warming is strongest), and Fig. 3c shows the increase in PBLH is focused in the daytime, which is also when the BC concentration is reduced. On average, however, as shown in Fig. 2c, this daytime response dominates, causing an increase in CO₂ to be associated with a reduction in near-surface BC. We have added a couple of sentences (bold text below) in the description of Fig. 3 where we clearly state this, in order to make these diurnal differences clear to the reader:

“The increased turbulence causes an increase in PBL height that is statistically significant at the same time of day that we see a reduction in near-surface BC particles (Fig. 3c). **These changes are not as strong as for a tenfold increase in BC, and changes to atmospheric stability, turbulence, PBLH and near-surface aerosol concentration are slightly opposite during night-time. Still, on average, as seen in Fig. 2c, increased CO₂ leads to a reduction in near-surface aerosols.**”

2. The model resolution is too coarse to resolve any boundary layer process. CESM only has 2-3 levels within the PBLH, whereas to resolve turbulence, it typically requires vertical resolutions less than 50m. Therefore, this model can be used to diagnose convective stability at best, and the PBLH indicated is only convective boundary layer height. The results do not reflect any information of turbulence.

A global climate model is indeed too coarse to resolve the boundary layer processes. However, the lowermost levels of CAM6 in the daytime and in the summer) do still give us an estimate of the shear, and enough information to provide the bulk Richardson’s number Ri. Thus, we are able to say something about the probability of turbulence, and the changes in that probability. In order to investigate whether the CAM6 changes in boundary layer height (which is calculated based on Ri in the model) would have looked different if the resolution had been higher, we have chosen to perform a new set of simulations using the regional climate model WRF. WRF has a higher horizontal as well as vertical resolution – still not on the LES level necessary to resolve turbulence, but with about 10 layers in the boundary layer and a resolution of less than 50 m in the lowest three model layers. Granted, this is another model and not directly comparable (see a more thorough description in the Methods section of the manuscript), but we believe the strong perturbations we perform should give comparable responses if the responses are robust.

We have added new figures S2 and S3 to the Supplementary. Firstly, we find that the boundary layer height over the Eastern China region, while expectedly not identical between the two models, is comparable and in the vicinity of 700 m on average:

We have also looked at the diurnal cycle in PBL height to see if CESM-CAM6 captures the most important features of the growing/narrowing boundary layer over the diurnal cycle compared to a higher-resolution model. In general the evolution of the PBL height between the CESM and the WRF models is very similar, as is the PBL height between the two different WRF resolutions. The CESM

model seems to be a bit slower in developing the daytime PBL, and reaches slightly higher levels, and the highest WRF resolution has a slightly shallower PBL than the lower WRF resolution (leftmost panel below). The influence of BC on the PBL height is very similar between the two WRF versions and the CESM (rightmost panel below).

The response of the PBL height to BC is also very similar – both in regional means but also in geographical distribution in spite of the fact that CESM grid cells are close to 100x100 km at these latitudes.

A discussion of these new simulations is now added after the discussion of Fig. 1: “Choosing a global climate model as a tool for this study has the advantage of global spatial coverage and hundred years of temporal coverage – thus allowing for an averaging over many different seasons and meteorological background conditions. However, as mentioned, the coarse horizontal and vertical resolution means that boundary layer processes such as turbulence are parametrized rather than resolved. To investigate the consequence of model resolution to the overall response seen in Fig. 1, we perform an additional set of corresponding idealized simulations using the regional climate model WRF (see Methods), downscaled from the CESM2 fixed sea-surface temperature simulations. The WRF simulations are run at 45 km and 15 km horizontal resolution for a region over Eastern China marked in Fig. 1a. While these are still not high enough resolutions to resolve turbulence, the vertical grid spacing in WRF is approximately 50 m in the lowest three model layers, which means that estimates of Ri and PBL changes should be more accurate. Figure S2 shows that the diurnal evolution of the PBL height in CESM2, which has a resolution of about 100 km at these latitudes, is very similar to the WRF simulations both in terms of baseline and changes in PBL height, albeit with a slightly delayed (1-2 hours) evolution of the daytime PBL and a weaker response to the sulfate perturbation compared to WRF. Geographical patterns in baseline PBL height and PBL height changes are also robust between CESM2 and WRF, as well as between the two resolutions of WRF (Fig. S3). In summary, the regional simulations indicate that the sensitivity of our results to model resolution is relatively small, at least for the range of scales studied here, giving confidence in our CESM2 results.”

3. The analysis is very simple and many conclusions are speculative. All quantitative conclusions are based on idealized and sometimes extreme cases such 10 times BC and 2 times CO₂.

The choice of performing simulations of idealized, strong emission increases, was motivated by the need for increasing the signal-to-noise ratio. An alternative approach could have been to perform multiple ensemble simulations of more realistic perturbations. We chose the former method as the simulations were computationally expensive and storage (recall that we have hourly output for 100 years for each of the simulations) was a challenge. We try to link these idealized simulations to the more realistic (in terms of emissions) future simulations from CMIP6, but from the reviewer's comment we understand that we need to be even more clear that we realize that these idealized simulations are unrealistic. We have therefore added the following to the end of the section describing these simulations:

"The highly idealized nature and the exaggerated magnitude of these perturbations, while necessary to produce strong enough signal-to-noise ratio to analyze these processes, preclude any direct linking between our findings and current trends in CO₂, BC or SO₄ in the real world. However, knowing how changes in these constituents will influence near-surface pollution levels can help us interpret more realistic simulations of future emission changes. In the next section, we take a closer look at such simulations from the Sixth Coupled Model Intercomparison Project (CMIP6)."

The change in surface pollutants is actually the change in BC, which also largely depends on the assumed emission. Also, the changes in other pollutants, such as secondary aerosols and gases, depends on many meteorological factors beyond PBLH. These limitations should be explicitly discussed.

Yes, the choice of near-surface BC concentration as a proxy for "surface pollutants" was driven mainly by limitations in model output. We do state this in the text and figure captions but should perhaps have been even more clear. We have therefore added a paragraph in the Methods section: "In this work we have chosen to use the concentration of black carbon as a proxy for "near surface aerosol concentrations". This is partly motivated by the fact that these particles are shown to have a particularly detrimental effect on health (Janssen et al., 2011), and partly necessitated by the limitation of hourly output from the model. BC concentration was available as output for hourly, daily and monthly temporal resolutions from all of the simulations. In reality, the perturbations performed in our simulations may influence the near-surface concentration of all types of pollutants. Changes in processes that only influence highly hydrophilic aerosols, or gaseous pollutants, will not be captured by our approach. However, our assumption is that the changes we see in planetary boundary layer height will cause a concentration or dilution of all types of particulate pollution and will therefore also be seen in changes in our BC output."

In addition, we have also added some text to comment on this caveat in the end of the paragraph discussing Figure 2:

"Thus, emissions in greenhouse gases as well as emissions of both absorbing and scattering aerosols seem to influence turbulence, PBL height and near-surface aerosol concentration. That does not mean, however, that it is turbulence and PBL height changes that cause the aerosol concentration changes in all cases. Many meteorological processes may influence the concentration of near-surface BC. For instance, an increase in precipitation will lead to increased aerosol wet deposition and reduced aerosol numbers, and a reduction in clouds may trigger changes in photolytically active particulate or gaseous pollutants. The nature of the simulations performed in this study does not allow for a clean separation into impacts from different meteorological effects. However, to get an indication of the processes in play and whether it is the boundary layer feedback that causes the links found above, we investigate the *timing* of the processes. We again focus on wintertime conditions over East China and take a closer look at changes in the regional mean diurnal cycles."

Also, the results largely depend on the vertical distribution of the forcing agents, especially aerosols, yet there is not description how these profiles are specified in the simulation, and whether they represent real conditions.

The vertical distribution of especially the absorbing aerosols is indeed of great importance, and we agree that we should have described the model BC profiles in the manuscript. In the figure below, we show baseline concentrations of BC for the surface (map) as well as vertical profiles, for different regions (marked by colors). The vertical profiles of aerosols are simulated based on real processes in the model.

There is a well-known tendency for the global climate models to underestimate the concentrations of near-surface levels of BC (Eckhardt et al., 2015), with a particular inability to capture the intense haze events. In that sense, the estimates of the number of high-pollution events shown in this paper is likely to have a low bias compared to the real world. Global climate models have also been shown to have a high bias in the upper atmosphere (Eckhardt et al., 2015). Both this high bias and the lower-atmosphere low bias have been shown to be at least partly linked to too efficient vertical transport in many models. A short discussion of these aspects has now been added to the manuscript, in the Methods section:

“The vertical distribution of BC, relevant to the distribution of atmospheric heating and thus to stability changes, is simulated by the model and depends on processes such as dry and wet deposition, advection, and vertical transport. In a thorough observational comparison with an earlier version of CAM, the model was found to capture the main features of the variation of BC with height, though with a tendency to overestimate BC concentrations in the upper troposphere, and underestimate observed BC concentrations at lower altitudes in high northern hemisphere latitudes (Liu et al., 2016). In China, the focus region of this study, models in general have been shown to have too low total column aerosol loads (Turnock et al., 2020) and too low surface aerosol concentrations (Xu et al., 2022). These are biases that are common to many global climate models (Eckhardt et al., 2015), and are at least partly linked to biases in wet deposition, vertical transport, and lacking emission sources. In the present study a low bias in the intense pollution events may cause us to underestimate the boundary layer effect of BC changes.”

Reviewer #2 (Comments to the Author):

This study focuses on the interactions between radiative forcers, specifically black carbon and sulphate aerosols and carbon dioxide (CO₂) and their potential to influence planetary boundary layer height through perturbing the temperature profile of the PBL and influencing turbulent motion. This study fits well and contributes to an increasing number of studies focusing on the aerosol-PBL feedback loop. This study is novel in the aspect of the use of a climate model rather than a smaller scale model which directly resolves turbulent motion. The advantage of using this type of model comes with the ability to assess the impact of various scenarios into the future to understand the

impact changes in emissions could have on pollution episodes. The manuscript is overall well written and the methodology is clear. General comments and suggestions to the authors are detailed below.

We thank the reviewer for good comments and a constructive review.

General Comments:

1) Throughout the manuscript it is stated that black carbon can only cause suppression in the PBL, through reducing solar radiation reaching the surface. However, this is dependent on the altitude of the BC layer. A few papers including Wang et. al 2018 (<https://doi.org/10.5194/acp-18-2821-2018>) and Slater et. al 2022 (<https://doi.org/10.5194/acp-22-2937-2022>) highlight the impact that the altitude of the BC layer can have on a) The magnitude of suppression caused by BC in the PBL and b) Whether it has the potential to promote turbulence e.g if a BC layer exists only at the surface and not throughout the atmospheric column. Whether these conditions occur often is uncertain but some studies have shown there can be variation in both concentrations and compositions of aerosols across the column, particularly in very polluted conditions. It may be worth adding a sentence or two to highlight this - both in relation to BC and SO₄.

This point is indeed under communicated in our manuscript. The dependency of the impact on the altitude of the BC particles has been well established since (Koch and Del Genio, 2010) and is discussed in several later papers including the excellent papers suggested by the reviewer. While our possibilities to investigate the altitudinal dependence of the boundary layer feedback to BC are very limited in our global climate model, we completely agree that this should be mentioned. We have chosen to rewrite one of the paragraphs in the Introduction (bold is new text):

“Recent findings indicate the existence of a feedback effect wherein high emissions cause reduced turbulence in the lower atmosphere. This boundary layer feedback effect can exacerbate episodes of severely enhanced pollution substantially (Petäjä et al., 2016; Qu et al., 2018; Wang et al., 2020; Wilcox et al., 2016). **Absorbing aerosols, such as black carbon (BC), initiates atmospheric heating through solar absorption and re-emission of long wave radiation. As shown in observational studies (e.g. (Wang et al., 2019) and (Slater et al., 2022)), the way the atmospheric heating influences the planetary boundary layer depends on the altitude of the particles. If the BC particles remain close to the surface, this added near-surface heating can increase convection and turbulence. However, the particles are often transported upwards in the atmosphere, and at higher altitudes an atmospheric heating reduces the boundary layer lapse rate, resulting in a shallower PBL.**”

2) An important output of this study could be the links between climate change mitigation (e.g reducing emissions of CO₂ and BC) and the additional benefits for reduced pollution and consequently human health beyond those which are typically cited. E.g reducing emissions of CO₂ and BC could have the added impact of allowing for more vertical mixing/turbulence and therefore have an additional effect (beyond reducing emissions) on human health through decreasing the number of severe pollution episodes (both in longevity and severity)

Yes, this is indeed the message that we aim for with this paper. We are however not entirely sure if the reviewer thinks that this message should have been conveyed in a different manner than what we do. We have therefore not done any large changes in this regard but remind the reviewer of our discussion of this topic in the section “Application to historical and future trends”. We have added a few words (bold text) in the abstract to make it even clearer: “Using the CMIP6 ensemble we find that in future scenarios with increasing CO₂ and reduced aerosol emissions, around 10% of the world’s population currently living in regions with high pollution levels are likely to experience a particularly strong increase in turbulence and PBL height, and thus a reduction in intense pollution

events. Our results clearly underline the **added positive impact** of BC mitigation for reducing the most severe exposures to air pollution **and the detrimental human health effects that follow.**"

3) Relating to point 1 - does the model simulation set up here assume uniform vertical aerosol profiles? It would be useful to state this somewhere and to highlight it as a potential limitation as well as further highlighting the limitations of using a coarse model to understand PBL turbulence.

Thank you, this is a good point. The vertical aerosol profiles in the model are not uniform, and we agree that this is an important point which needs to be discussed in the paper. Please see the final question (and our response) of Reviewer #1 for more details on the vertical profiles of BC in CAM6, and how they compare to observations.

We also agree that the limitations and caveats of using a coarse global climate model to understand PBL turbulence should be discussed more explicitly in the manuscript. We have now added a separate section on this in the manuscript, after the description of Fig. 1. We have performed additional simulations using a finely resolved regional climate model (WRF), to compare how changes in PBL height varies with model resolution. Figures from these simulations are added as supplementary information, but as an example the figure below demonstrates that the coarse CESM2-CAM6 show very similar changes in PBL height due to BC changes as WRF at both 45 and 15 km resolution:

In addition, we note that the diurnal cycle in PBL height as well as the diurnal variation in change in PBL height due to BC is similar between all three resolutions:

4) Line 145-150: It is stated that in Eastern China the PBL feedback effect is larger due to increased absorbing aerosols compared to in Southern China. However, there may be other factors influencing this - for example meteorological conditions which have been shown to impact this effect see Wang et. al 2019 (<https://doi.org/10.5194/acp-19-6949-2019>).

We absolutely agree, this sentence was a pure reference to another paper wherein they interpreted the East/South difference as driven by absorbing/scattering aerosols. Indeed, there are plenty other differences between these two regions to cause differences in PBL feedback – for instance meteorological conditions as suggested by the reviewer. We have added a paragraph to comment on this caveat in the end of the paragraph discussing Figure 2:

“Thus, emissions in greenhouse gases as well as emissions of both absorbing and scattering aerosols seem to influence turbulence, PBL height and near-surface aerosol concentration. That does not mean, however, that it is turbulence and PBL height changes that cause the aerosol concentration changes in all cases. Many meteorological processes may influence the concentration of near-surface BC (Wang et al., 2019). For instance, an increase in precipitation will lead to increased aerosol wet deposition and reduced aerosol numbers, and a reduction in clouds may trigger changes in photolytically active particulate or gaseous pollutants. The nature of the simulations performed in this study does not allow for a clean separation into impacts from different meteorological effects. However, to get an indication of the processes in play and whether it is the boundary layer feedback that causes the links found above, we investigate the *timing* of the processes. We again focus on wintertime conditions over East China and take a closer look at changes in the regional mean diurnal cycles.”

5) Does the model take into account the impacts of projected increases in surface temperature due to climate change? What impact does increased surface temperatures have on this effect?

It does indeed take surface temperature change into account, and this is a crucial part of the observed changes in PBLH in particular for the CO₂ perturbation. The added CO₂ causes, in time, an increase in surface temperatures which destabilizes the lower atmosphere and increases turbulence and boundary layer height. This is thoroughly discussed in the section describing Figure 3.

Throughout the manuscript it would be useful to clarify a few terms:

Line 24: Morbidity rates

Thank you, this is clarified now.

Line 34: What is meant by unfavourable meteorological conditions e.g prevailing winds, pressure systems, temperature, humidity etc.

Thank you, this is now clarified in the text.

Reviewer #3 (Comments to the Author):

Reviewer comments for the paper “The turbulent future – a breath of fresh air” by Stjern et al., submitted for publication in Nature Communications

The authors utilize data from global model and from CMIP6 ensemble to explore the impacts of feedback mechanics impacting the boundary layer height and turbulence. The driving force of these changes stem from changes in aerosol emissions and anticipated CO₂ concentrations in future climate. The noteworthy analysis includes the use of global data set from model ensemble explore the impacts of concurrent changes in aerosol emissions and warming climate and turbulence. The authors depict a convincing but a hopeful scenario that could improve air quality in rapidly urbanizing environment in the coming decades, aided by strong air quality controls that would also improve the health of the urban population.

As a summary, the work a relevant contribution providing novel insights into aerosol-boundary layer

feedbacks. Therefore, I recommend to publish the paper after considering my comments that I summarize in the comments below.

We thank the reviewer for this thorough review with constructive comments. We hope that we have managed to address these comments in a satisfactory manner – please see blue text below your comments in black.

Significance of the work to the field, originality

The authors utilize a global data in their analysis of impacts changing emissions and boundary layer height and turbulence. As the authors point out, these feedback mechanisms are typically only addressed in short experiments and not with in a global context in the changing climate. This is the novelty of the work.

The authors differentiate absorbing aerosols and scattering aerosols in their analysis in a common framework. This is an important step forward in the analysis of boundary layer – aerosol feedback analysis.

Solidity of the conclusions and claims, any additional evidence needed?

However, since I'm not a global modeler, I cannot address fully the capacity of the models to perform well enough for such endeavors. I list below few concerns that should be addressed and clarified: 1) The simulations with CESM2-CAM6 are performed with doubled CO₂ and separate model runs with tenfold increase in BC, and fivefold increase in SO₄. Is this done uniformly for the global emissions with keeping the regional differences?

Yes it is! This is now clarified in the text.

Are these very high increases made to enhance the anticipated signal?

That is correct. We have added the following comment to clarify this in the Methods section of the manuscript:

“The magnitudes of the experiments are artificially strong compared to real-world emission changes in order to produce clear enough signals to more easily interpret the resulting atmospheric process changes.”

Are there any unwanted consequences (or interactions) coming from the perturbations? How does e.g the surface coating of BC due to sulfuric acid condensation influence the overall picture?

The reviewer poses a valid question, there are indeed unwanted interactions between the BC and SO₄ perturbations. Specifically, when perturbing SO₄ we inadvertently boost the coating of BC and so added SO₄ emissions cause changes in both types of aerosols. This makes the effect of SO₄ aerosols alone difficult to extract, and we have therefore chosen caution in our formulations of the influence of SO₄ on turbulence throughout the manuscript. For instance, the description of the SO₄ response in Fig. 3 currently reads:

“While there is a significant increase in near-surface BC, shown also in Fig. 2c, in this case this is clearly not driven by the boundary layer feedback. Instead, by looking at changes in the model's different aerosol modes (the primary carbon mode and the BC accumulation mode), we find that the added SO₄ causes a very strong reduction in primary mode and increase in accumulation mode BC. In the CAM6 MAM4 aerosol scheme, BC particles “grow” through coating by SO₄ or SOA, and this constitutes the strong removal of primary mode and addition of accumulation mode BC aerosols. This model response, although approximating a similar effect in nature, makes it difficult to draw conclusions as to the boundary layer feedback effect of SO₄ from these simulations. In a future study we will use a regional model to isolate the link more properly between the stabilizing effect of SO₄ and surface pollution, comparing the effects of SO₄ and BC.”

How reasonable is the tenfold and fivefold increase in BC and SO₄ emissions? On one hand, according to Xu et al. (2021) the BC has approximately doubled from 4 Tg to 8 Tg in 57 years. 10-fold increase is large in this context. On the other hand, global sulfur (as SO₂ and as SO₄) emissions are declining (Aas et al. 2019).

This is a good question, and one that many readers are likely to ask. These perturbations are not “reasonable” in terms of comparability to for instance preindustrial to present-day emission changes. As mentioned above, we expect signals to be relatively small compared to natural variability. To remedy this, we had the choice between performing many ensemble runs and doing exaggerated perturbations, and we chose the latter as the first option was not feasible due to vast computational requirements. The specific perturbation used here has been shown in earlier model studies to give a good signal-to-noise ratio (Myhre et al., 2017). We try to link these idealized simulations to the more realistic (in terms of emissions) future simulations from CMIP6, but from the reviewer’s comment we understand that we need to be even more clear that we realize that these idealized simulations are unrealistic. We have therefore added the following to the end of the section describing these simulations:

“The highly idealized nature and the exaggerated magnitude of these perturbations, while necessary to produce strong enough signal-to-noise ratio to analyze these processes, preclude any direct linking between our findings and current trends in CO₂, BC or SO₄ in the real world. However, knowing how changes in these constituents will influence near-surface pollution levels can help us interpret more realistic simulations of future emission changes.”

2) Is the vertical resolution of the model enough to capture the full dynamics of the growing boundary layer during its diurnal cycle?

The boundary layer over China typically covers between 2 and 7 CAM6 model layers, and the fine scale variations in surface fluxes and their vertical distribution cannot fully be captured. However, the model *does* simulate the basic diurnal variations in radiative and surface fluxes and therefore the basic diurnal cycle in PBL height, with a narrower boundary layer in the nighttime and a maximum PBL height in the afternoon. In order to compare PBLH changes between CAM6 and a higher-resolution (both in the horizontal and the vertical) regional model, we have now performed additional simulations using the regional climate model WRF. These results will be shown below, in response to the reviewer’s question on whether CAM6 turbulence changes can be tested against higher resolution models. Here, however, we show the diurnal PBLH cycle in CAM6 (spatial resolution of around 100km at latitudes around the China region) as well as in WRF at both 45km and 15 km resolution:

We see that the diurnal PBLH cycle is relatively similar in CAM6 to even the highest resolution WRF runs, despite the coarse resolution of the global climate model.

Finally, we also stress that even if the climatology of the diurnal boundary layer cycle in CAM6 is not captured perfectly, our focus in this analysis is on the *change* in PBLH, which we argue would influence boundary layer aerosol concentration regardless.

What are the consequences of using Ri as an indication for the turbulence?

Actual turbulence is of course not possible to derive in a global climate model, and Ri is chosen as the turbulence metric in this paper because it is a well-established and one of the most common proxies for turbulence – or, rather, for the probability for the onset of turbulence. It is used as such a proxy in both models and observations alike, and thus the more pressing concern is not the consequences of using Ri as a proxy for turbulence, but how models perform in calculating this parameter compared to observations.

Other turbulence-related variables could have been chosen for this study, such as the buoyancy flux (W/m^2 , left panel below) which is one of the parameters calculated by the model. As seen below, there are some differences in the pattern of change of these to turbulent quantities in response to an increase in BC, but in general patterns are very similar. In particular, the region of East China can be seen to experience a strong reduction in turbulent conditions in both quantities.

An advantage of using the Richardson’s number, however, is its clear link to the planetary boundary layer. Ri is often used to calculating PBL height in observations and models alike. As mentioned in the methods chapter, PBL height is a function of Ri in the CAM6 model. Seidel et al. (2012) compare boundary layer heights over Europe and USA calculated based on Ri measured both in radiosonde observations, in reanalyses, and in two different global climate models. One of the models they look at is the CAM5 model – the pre-runner to CAM6. The differences they find in PBL height thus originate from differences between modelled and observed Ri. In general, they find that the climate models simulate a higher nighttime PBLH than the radiosonde measurements, suggesting a general model difficulty in simulating shallow stable night-time boundary layers. CAM5 shows a smaller but positive bias in daytime PBLH (see extracts from their figures 4 and 8 below).

We have now added the following sentences to the methods section:

“We note, as a caveat, that global climate models in general struggle to simulate very shallow stable

boundary layers (Medeiros et al., 2011; Seidel et al., 2012). In this analysis, this tendency may lead to underestimates of the tendency of the boundary layer to narrow in response to increased BC.”

Can these be tested against observations or against higher resolution models?

The PBL height cannot be directly measured, and as such even observations rely on estimates. Several different methods exist, but the estimates based on Ri are common. As discussed above, there are indeed differences between observations and models in terms of Ri-based PBLH. The present analysis is based on highly idealized simulations and therefore we cannot compare our results to observations. However, we can investigate the consequences of spatial resolution on our results. In the figure below, we first compare PBLH in the baseline simulations of CAM6 (leftmost panel) to simulations using the regional climate model WRF. CAM6 has a spatial resolution (around the latitudes of the China region) of around 100 km. The middle panel shows PLBH from a simulation with WRF using the AOD fields from CAM6 as input to the radiation scheme and a spatial resolution of 45 km. The rightmost panel shows a WRF simulation with spatial resolution of 15 km.

In general, the PBLH from the higher-resolution simulations are similar to that from the global climate model, albeit with a tendency for the latter to simulate higher PBLH over the more mountainous regions in the northwestern part of the domain.

We have also performed the idealized perturbation experiments with WRF, and below are maps showing PBLH changes for BCx10. Compared to the higher-resolution runs, we see that CAM6 is in fact able to simulate the broad strokes of the general PBLH changes quite well.

3) How about horizontal resolution? How are the differences in local anthropogenic heating (urban heat island) taken into account in the development of turbulent in urban environments?

The global climate model will indeed not be able to capture urban heat island effects and associated turbulent changes and thus this effect is not taken into account in our simulations. Although this is a mechanism missing in our simulations, our scope is to investigate the more general impact of the perturbations in anthropogenic emissions, for which our global climate model is better fit.

Flaws in the data analysis, interpretation and conclusions. Revisions needed?

I don't see any flaws in the analysis or interpretation of the results. However, see comments above

regarding the model resolution and comment on the perturbations.

See comments above, regarding both model resolution and the magnitude of the perturbations. We believe the conclusions in this work are justified.

Soundness of methodology, does it meet the expectations

The methodology fits the work, given that the model spatial and temporal resolution are adequate. The work provides as a step forward in global context by expanding the process level studies and regional studies referred in the paper.

Thanks you for these positive comments.

Enough details provided?

Generally yes, but technical details regarding the modeling should be elaborated in more detail.

We have now improved our description of the model and the experiment set-up as suggested by you and the other reviewers.

Detailed comments.

The paper is very well written. For the internal structure, I actually got more interested in the topic with the detailed analysis and Figure 2. The story in the paper could improve, if it was approached from the details into the global picture.

Thank you for this comment. We understand the reviewer's suggestion and in fact, in an earlier version of the manuscript we organized the paper exactly as suggested. However, we struggled with the flow of the text when organized this way, and therefore – while the idea is excellent – we choose to keep the current structure.

Line 252-253: Can you quantify the reduction of intense haze events?

That would indeed have been an excellent addition this analysis, and one we initially hoped to make. However, these climate models do not have daily surface aerosol concentration as standard output, and thus we are left with looking at changes in boundary layer height and just point back at the link we establish between changes in boundary layer height and changes in near-surface aerosol pollution in our idealized simulations.

In superlatives, please use “the” in proper places.

Thank you, we have gone through the manuscript and fixed this in several places.

References

Aas et al. (2019) Global and regional trends of atmospheric sulfur, *Scientific Reports* 9, 953.

Xu et al. (2021) Updated Global Black Carbon Emissions from 1960 to 2017: Improvements, Trends, and Drivers, *Environ. Sci. Technol.* 2021, 55, 12, 7869–7879.

Eckhardt, S., Quennehen, B., Olivie, D. J. L., Berntsen, T. K., Cherian, R., Christensen, J. H., Collins, W., Crepinsek, S., Daskalakis, N., Flanner, M., Herber, A., Heyes, C., Hodnebrog, Ø., Huang, L., Kanakidou, M., Klimont, Z., Langner, J., Law, K. S., Lund, M. T., Mahmood, R., Massling, A., Myriokefalitakis, S., Nielsen, I. E., Nøjgaard, J. K., Quaas, J., Quinn, P. K., Raut, J. C., Rumbold, S. T., Schulz, M., Sharma, S., Skeie, R. B., Skov, H., Uttal, T., von Salzen, K., and Stohl, A.: Current model capabilities for simulating black carbon and sulfate concentrations in the Arctic atmosphere: a multi-model evaluation using a comprehensive measurement data set, *Atmos. Chem. Phys.*, 15, 9413-9433, 10.5194/acp-15-9413-2015, 2015.

Janssen, N. A., Hoek, G., Simic-Lawson, M., Fischer, P., van Bree, L., ten Brink, H., Keuken, M., Atkinson, R. W., Anderson, H. R., Brunekreef, B., and Cassee, F. R.: Black carbon as an additional indicator of the adverse health effects of airborne particles compared with PM10 and PM2.5, *Environ Health Perspect*, 119, 1691-1699, 10.1289/ehp.1003369, 2011.

Koch, D., and Del Genio, A. D.: Black carbon semi-direct effects on cloud cover: review and synthesis, *Atmos. Chem. Phys.*, 10, 7685-7696, 10.5194/acp-10-7685-2010, 2010.

Liu, X., Ma, P. L., Wang, H., Tilmes, S., Singh, B., Easter, R. C., Ghan, S. J., and Rasch, P. J.: Description and evaluation of a new four-mode version of the Modal Aerosol Module (MAM4) within version 5.3 of the Community Atmosphere Model, *Geosci. Model Dev.*, 9, 505-522, 10.5194/gmd-9-505-2016, 2016.

Medeiros, B., Deser, C., Tomas, R. A., and Kay, J. E.: Arctic Inversion Strength in Climate Models, *Journal of Climate*, 24, 4733-4740, 10.1175/2011jcli3968.1, 2011.

Myhre, G., Forster, P. M., Samset, B. H., Hodnebrog, Ø., Sillmann, J., Aalbergstjø, S. G., Andrews, T., Boucher, O., Faluvegi, G., Fläschner, D., Kasoar, M., Kharin, V., Kirkevåg, A., Lamarque, J.-F., Olivie, D., Richardson, T., Shindell, D., Shine, K. P., Stjern, C. W., Takemura, T., Voulgarakis, A., and Zwiers, F.: PDRMIP: A Precipitation Driver and Response Model Intercomparison Project, Protocol and preliminary results, *Bulletin of the American Meteorological Society*, 98, 1185-1198, doi: 10.1175/BAMS-D-16-0019.1, 2017.

Petäjä, T., Järvi, L., Kerminen, V. M., Ding, A. J., Sun, J. N., Nie, W., Kujansuu, J., Virkkula, A., Yang, X., Fu, C. B., Zilitinkevich, S., and Kulmala, M.: Enhanced air pollution via aerosol-boundary layer feedback in China, *Scientific Reports*, 6, 18998, 10.1038/srep18998, 2016.

Qu, W., Wang, J., Zhang, X., Wang, Y., Gao, S., Zhao, C., Sun, L., Zhou, Y., Wang, W., Liu, X., Hu, H., and Huang, F.: Effect of weakened diurnal evolution of atmospheric boundary layer to air pollution over eastern China associated to aerosol, cloud – ABL feedback, *Atmospheric Environment*, 185, 168-179, <https://doi.org/10.1016/j.atmosenv.2018.05.014>, 2018.

Seidel, D. J., Zhang, Y., Beljaars, A., Golaz, J.-C., Jacobson, A. R., and Medeiros, B.: Climatology of the planetary boundary layer over the continental United States and Europe, *Journal of Geophysical Research: Atmospheres*, 117, <https://doi.org/10.1029/2012JD018143>, 2012.

Senf, F., Quaas, J., and Tegen, I.: Absorbing aerosol decreases cloud cover in cloud-resolving simulations over Germany, *Quarterly Journal of the Royal Meteorological Society*, 147, 4083-4100, <https://doi.org/10.1002/qj.4169>, 2021.

Slater, J., Coe, H., McFiggans, G., Tonttila, J., and Romakkaniemi, S.: The effect of BC on aerosol-boundary layer feedback: potential implications for urban pollution episodes, *Atmos. Chem. Phys.*, 22, 2937-2953, 10.5194/acp-22-2937-2022, 2022.

Turnock, S. T., Allen, R. J., Andrews, M., Bauer, S. E., Deushi, M., Emmons, L., Good, P., Horowitz, L., John, J. G., Michou, M., Nabat, P., Naik, V., Neubauer, D., O'Connor, F. M., Olivie, D., Oshima, N., Schulz, M., Sellar, A., Shim, S., Takemura, T., Tilmes, S., Tsigaridis, K., Wu, T., and Zhang, J.: Historical and future changes in air pollutants from CMIP6 models, *Atmos. Chem. Phys.*, 20, 14547-14579, 10.5194/acp-20-14547-2020, 2020.

Wang, L., Liu, J., Gao, Z., Li, Y., Huang, M., Fan, S., Zhang, X., Yang, Y., Miao, S., Zou, H., Sun, Y., Chen, Y., and Yang, T.: Vertical observations of the atmospheric boundary layer structure over Beijing urban area during air pollution episodes, *Atmos. Chem. Phys.*, 19, 6949-6967, 10.5194/acp-19-6949-2019, 2019.

Wang, Y., Yu, M., Wang, Y., Tang, G., Song, T., Zhou, P., Liu, Z., Hu, B., Ji, D., Wang, L., Zhu, X., Yan, C., Ehn, M., Gao, W., Pan, Y., Xin, J., Sun, Y., Kerminen, V. M., Kulmala, M., and Petäjä, T.: Rapid formation of intense haze episodes via aerosol-boundary layer feedback in Beijing, *Atmos. Chem. Phys.*, 20, 45-53, 10.5194/acp-20-45-2020, 2020.

Wilcox, E. M., Thomas, R. M., Praveen, P. S., Pistone, K., Bender, F. A.-M., and Ramanathan, V.: Black carbon solar absorption suppresses turbulence in the atmospheric boundary layer, *Proceedings of the National Academy of Sciences*, 113, 11794-11799, 10.1073/pnas.1525746113, 2016.

Xu, Y., Wu, J., and Han, Z.: Evaluation and Projection of Surface PM_{2.5} and Its Exposure on Population in Asia Based on the CMIP6 GCMs, *International Journal of Environmental Research and Public Health*, 19, 12092, 2022.

REVIEWER COMMENTS

Reviewer #1 (Remarks to the Author):

The authors have addressed some of my comments and the manuscript is improved. However, I find the study still needs clarifications.

Specific comments:

1. The authors argue that their focus is on global perspective rather than regional, which proves the novelty of their study. I cannot agree with this. PBLH is a local parameter, depending on the local vertical profile. The "global results" are just a sum of all local results. And on local scale, these effects are already known. The authors also picked China as an example and discussed in more detail, however, there are already numerous works on PBL-aerosol feedback focusing on the China region. Pollution is also mostly a regional phenomenon and the reduction of PBLH mainly matters where pollution is heavy. It's not appropriate to discuss pollution reduction on global scale, since it depends on local emission and meteorology.
2. The authors claimed that on global scale, turbulence will increase in the future. However, based on Figure 1, PBLH decreased over the oceans in the 2xCO₂ scenario. In the aerosol scenario, PBLH also changed oppositely between ocean and land. So if we calculate a global mean turbulence and PBLH, we will end up a less turbulent and lower PBLH. The change of ocean PBLH, however, is not discussed.
3. There is no indication of the significance of future PBLH and Ri changes, such as Figure 1 and 3. Also, it is more informative to present the changes in percentage rather than absolute number, since the PBLH and turbulence themselves have regional differences. The significance of the correlation in Figure 2 is also not given.
4. The authors argued a "breath of fresh air" in the future. However, they did not prove whether the change in PBLH make a significant contribution to future air pollution reduction. As projected by CMIP6, emission of pollutants will decrease significantly in the future, which is the cause of the PBLH increase. So how much more reduction will the PBLH increase add? How important is it compared to emission reduction itself? If emission is already so low to cause any serious pollution, do we need to care the changes in PBLH, in the pollution perspective?

Reviewer #3 (Remarks to the Author):

As I stated in my original review, the manuscript is very relevant and interesting. It is a relevant contribution providing novel insights into aerosol-boundary layer feedbacks. In my opinion, the authors have answered my concerns about the paper. Therefore, I recommend the paper to be published.

Response to reviewers

“The turbulent future – a breath of fresh air”, by Stjern et al.

Reviewer #1 (Remarks to the Author):

The authors have addressed some of my comments and the manuscript is improved. However, I find the study still needs clarifications.

Specific comments:

1. The authors argue that their focus is on global perspective rather than regional, which proves the novelty of their study. I cannot agree with this. PBLH is a local parameter, depending on the local vertical profile. The "global results" are just a sum of all local results. And on local scale, these effects are already known.

We understand the reviewer's point about PBLH being a highly local parameter. However, with a “global perspective” we mean that our tool (the global climate model) allows us to show global maps of changes in PBLH and related parameter for the entire globe (Fig. 1). We do not mean that the strength of this study is that we present global mean results. The only place in the paper where changes over all global land grid regions are shown in a sort of aggregated manner is in Fig. 2a, where we show scatterplots of changes in PBLH versus Ri in all land grid cells. Here, however, the “local” relationship between the two variables is kept intact and not averaged out.

While local relationships between pollution and turbulence/PBLH are to a large extent known, the clean comparison between how CO₂, BC and SO₄ influence these relationships is new.

We have now looked through all occurrences of the word “global” in the manuscript and we found two places where the reader might be misled to believe that we are presenting global mean results. We have chosen to change the text in two places:

1. In the beginning of the fifth paragraph of the introduction, we change “In this paper, we investigate the effect of individual climate drivers on the boundary layer feedback, in global as well as a regional perspective, and with a look into the future.” to “In this paper, we investigate the effect of individual climate drivers on the boundary layer feedback, focusing on land regions.”

2. In the final paragraph of the introduction, we change “Our focus is on responses over global land regions...” to “Our focus is on responses over land...”

The authors also picked China as an example and discussed in more detail, however, there are already numerous works on PBL-aerosol feedback focusing on the China region.

Granted, we have chosen a region for which much has already been shown and published. However, the majority of existing studies focusing on this region have been shorter “case” studies, based either on a few years of ground observations or on shorter observational campaigns. Regional model studies have also been limited in time. We believe that, given the caveats that we now discuss more thoroughly in the manuscript, the ability to use 100-year simulations incorporating a large range of different weather situations is an interesting addition to existing studies. These simulations have the added advantage to illustrate both the fast and the slow climate responses. Finally, we again want to stress the comparison between the three climate drivers, which even for this region have not been shown before.

Pollution is also mostly a regional phenomenon and the reduction of PBLH mainly matters where pollution is heavy. It's not appropriate to discuss pollution reduction on global scale, since it depends on local emission and meteorology.

We appreciate that pollution, as well as many other parameters of relevance to human health (extreme heat or precipitation, for instance), is strongly dependent on regional conditions such as emissions and meteorology. We do not believe, however, that this renders it meaningless to aggregate results – for instance by comparing how many people will feel the detrimental effects of pollution now versus in the future.

2. The authors claimed that on global scale, turbulence will increase in the future. However, based on Figure 1, PBLH decreased over the oceans in the 2xCO₂ scenario. In the aerosol scenario, PBLH also changed oppositely between ocean and land. So if we calculate a global mean turbulence and PBLH, we will end up a less turbulent and lower PBLH. The change of ocean PBLH, however, is not discussed.

Yes, there are indeed strong differences between responses to CO₂ (and the same can be seen for SO₄ in fig. 1) over land and over ocean, as the reviewer notes. However, our choice in this study is to focus on land regions as this is where people live.

In the introduction, we have the following sentences to specify our land focus:

“In this paper, we investigate the effect of individual climate drivers on the boundary layer feedback, focusing on land regions”

“Our focus is on responses over land, as aerosols over ocean will have little health impacts on people.”

The second section of RESULTS reads

“For the BC perturbation, we see a particularly strong increase in stability and reduction in turbulence and PBL height in the regions of high aerosol emissions. For the SO₄ perturbation there is also a general (but weaker) reduction in turbulence over land regions, while the CO₂ doubling causes an opposite reduction in stability and increase in turbulence and PBL height over land.”

..and in the third section we begin a discussion of land/ocean contrasts in results with the following sentence:

“While our focus will be on changes over land, we still note that the BC perturbation causes similar changes to turbulence and PBL height over land and ocean, while changes to SO₄ emissions and CO₂ concentrations cause opposite changes over land and ocean.”

Given this already existing text, we believe that the reader understands that our results have indeed a focus on land regions. We do discuss the PBLH change over ocean, contrary to what the reviewer states, although this is indeed only summarized in one paragraph since it is not the focus of this analysis.

3. There is no indication of the significance of future PBLH and Ri changes, such as Figure 1 and 3.

In Figure 3, we do indicate significance the text and in panels c also in the figure. See, for instance this text describing the figure:

“Through reduced surface fluxes, this leads to an increase in the Richardson number, strongest around 15.00 hours (Fig. 3b) but statistically significant by Student’s t-test for all hours of the day. The diurnal variation in the BC-induced temperature increases and turbulence reduction connects closely to a change in PBL height (black line in Fig. 2c), where hours with significant change are

marked with star symbols. In the BC perturbation, the PBL height change is significant for all hours of the day but strongest at 15.00 hours.”

As for Fig. 1, however, there should indeed have been similar hatchings as in Fig. 4. This is now fixed.

Also, it is more informative to present the changes in percentage rather than absolute number, since the PBLH and turbulence themselves have regional differences.

We did indeed show relative changes in percentage in an early version of Figure 1. However, we ended up with the absolute changes shown in the current version because the relative changes were dominated by regions where baseline values were very small and thus where relative changes were very high. We appreciate the idea, but we believe the current version is the most informative.

The significance of the correlation in Figure 2 is also not given.

The p-value is very close to zero for all three correlations in Fig. 2a, which is why we did not print the p-value on the panel, but we do state in the figure caption that “all correlation coefficients are statistically significant at the 0.01 level”.

4. The authors argued a "breath of fresh air" in the future. However, they did not prove whether the change in PBLH make a significant contribution to future air pollution reduction. As projected by CMIP6, emission of pollutants will decrease significantly in the future, which is the cause of the PBLH increase. So how much more reduction will the PBLH increase add? How important is it compared to emission reduction itself? If emission is already so low to cause any serious pollution, do we need to care the changes in PBLH, in the pollution perspective?

Thank you, this is an important point and one that should be clarified in the manuscript. Ideally, we would have had the possibility to perform simulations where the PBLH height had been held fixed, allowing for a quantification of the impact of emission changes versus PBLH changes on near-surface pollution levels. This is not possible in our model, however. Also, as we mention in the manuscript, projected future values of near-surface aerosols were not available to us for the CMIP6 data. In the final section of the manuscript, we therefore do not attempt any quantification as to future near-surface pollution levels. Instead, we discuss future PBLH changes for different scenarios and set them into context of the aerosol-PBLH relationships found from the idealized experiments of CAM6, for which at least historical PBLH trends are very similar to that of the CMIP6 ensemble mean (Fig. S4).

In the idealized experiments we show how near-surface BC concentrations (Fig. 2b) display much more than a tenfold increase following a tenfold enhancement of emissions. Following future aerosol emission reductions, we believe this non-linearity might provide an *added* positive influence on near-surface pollution levels. Of course, such a bonus effect is of limited value in the far future where (hopefully) pollution levels are very low. However, in many regions quite large emission reductions are needed to move below that limit, and even then, there will be exposure during haze episodes. Another point is that especially near-term emission levels are highly uncertain, as seen for instance by the highly contrasting BC emission evolutions for SSP370 and SSP585 in Fig. 4b. It is not given that emission reductions will commence until after a few decades.

The red text is added to the third paragraph of the last section, discussing future changes: “Given the tendency of BC to suppress turbulence and CO₂ to enhance turbulence, as found in our idealized model simulations, both an increase in CO₂ concentration and a reduction in BC concentrations will contribute to enhanced turbulence, with the potential positive consequence of an alleviation in the number of haze events. **Note that while the future reductions in BC emissions themselves will of course be the main determinant of health-related exposure to these aerosols, we are here interested in the added positive or negative influence by an associated change in PBL height.** We now investigate the link between emissions and PBL height in future scenarios, to get insight into

whether the boundary-layer feedback could have detrimental or advantageous impacts on near-surface pollution in the future.”

In addition, we have added this text as a final paragraph in the manuscript:

“Hopefully, the problems of aerosol pollution will eventually be alleviated in all regions of the globe, at which point the added impact of changes in turbulence is of little importance. In the meantime, air pollution is estimated to cause millions of deaths every year (WHO, 2021), and mitigations in the future do not necessarily provide instant relief as the number of people exposed to pollution-driven health risk is a function not only of emission levels, but also of demographic and socioeconomic factors. For instance, Chowdhury et al. (2018) find that excess deaths from PM_{2.5} in India are expected to increase towards 2100, due to demographic transitions involving population growth as well as a shift towards an older population. Ultimately, exposure to air pollution depends on a myriad of factors and is likely to continue to be an issue even as emissions go down. The way anthropogenic emissions interact with turbulence and feedback on the level of near-surface air pollution is one such factor, and one that is likely to contribute with a positive sign in the future.”

Reviewer #3 (Remarks to the Author):

As I stated in my original review, the manuscript is very relevant and interesting. It is a relevant contribution providing novel insights into aerosol-boundary layer feedbacks. In my opinion, the authors have answered my concerns about the paper. Therefore, I recommend the paper to be published.

Chowdhury, S., Dey, S., and Smith, K. R.: Ambient PM_{2.5} exposure and expected premature mortality to 2100 in India under climate change scenarios, *Nature Communications*, 9, 318, 10.1038/s41467-017-02755-y, 2018.

Ambient (outdoor) air pollution: [https://www.who.int/news-room/fact-sheets/detail/ambient-\(outdoor\)-air-quality-and-health](https://www.who.int/news-room/fact-sheets/detail/ambient-(outdoor)-air-quality-and-health), access: 10.13., 2021.